

# Evaluation of JULES-crop performance against site observations of irrigated maize from Mead, Nebraska

Karina Williams[1], Jemma Gornall[1], Anna Harper[2], Andy Wiltshire[1], Debbie Hemming[1], Tristan Quaife[3], Tim Arkebauer[4], and David Scoby[4]

[1]Met Office Hadley Centre, Exeter, UK
[2]College of Engineering, Mathematics, and Physical Sciences, University of Exeter, Exeter, UK
[3]National Centre for Earth Observation, Department of Meteorology, University of Reading, UK.
[4]Department of Agronomy and Horticulture, University of Nebraska-Lincoln, Lincoln, USA

*Correspondence to:* Karina Williams (karina.williams@metoffice.gov.uk)

**Abstract.**

The JULES-crop model (Osborne et al., 2015) is a parameterisation of crops within the Joint UK Land Environment Simulator (JULES), which aims to simulate both the impact of weather and climate on crop productivity and the impact of crop-lands on weather and climate. In this evaluation paper, observations of maize at three FLUXNET sites in Nebraska (US-Ne1, US-Ne2, US-Ne3) are used to test model assumptions and make appropriate input parameter choices. JULES runs are performed for the irrigated sites (US-Ne1 and US-Ne2) both with the crop model switched off (prescribing leaf area index (LAI) and canopy height) and with the crop model switched on. These are compared against GPP and carbon pool FLUXNET observations. We use the results to point to future priorities for model development and describe how our methodology can be adapted to set up model runs for other sites and crop varieties. The implications of our results on the choice of parameters and settings to be used in global runs of JULES-crop are also discussed.

## 1 Introduction

The Joint UK Land Environment Simulator (JULES) (Best et al., 2011; Clark et al., 2011) is a process-based model that simulates the fluxes of carbon, water, energy and momentum between the land surface and the atmosphere. It is used in carbon cycle, climate change and impacts studies, and can be run on its own ('standalone' mode) or as a component of a coupled Earth system model. As described in the model description paper Osborne et al. (2015), JULES-crop is a parameterisation of crops that has been added to JULES in order to improve land-atmosphere interactions in areas where crops predominate in addition to enabling the simulation of the effect of weather and climate on food and water resources.

JULES treats each vegetation type as existing on a separate tile within a gridbox. Energy and carbon flux calculations are performed separately for each tile and prognostics such as leaf area index (LAI) and canopy height are calculated and stored for each tile separately. Each vegetation tile has a different set of input parameters and leaf-level carbon assimilation is calculated differently depending on whether the tile is modelling a plant with a C3 or a C4 plant photosynthetic pathway. JULES-crop introduces a distinction between natural plant functional types (PFT) and crops. Crop tiles have their growth and development





parametrised by a crop development index (DVI) and have different calculations for the allocation to plant carbon pools, leaf area index and height compared to natural PFTs. However, in most other respects, such as the calculation of gross primary productivity (GPP) and respiration, natural PFTs and crops are modelled in the same way within the JULES code. In its current stage of implementation, JULES-crop is available only in offline JULES runs, although there are plans to extend it for use in coupled runs in the future.

In Osborne et al. (2015), global runs of the model were carried out for four generic crop types - maize, soybean, wheat and rice - and the effect of including the new crop parameterisation was shown on sensible heat flux, moisture flux and net primary productivity (NPP) for some key countries. The model yield was also compared against global and country FAO crop yields. Site runs were performed at four FLUXNET sites with maize-soybean rotation: Mead (US-Ne2 and US-Ne3), Bondville (US-Bo1) and Fermi (US-IB1). For input parameters which applied to both natural vegetation and crop tiles, C3 crops were given the parameter values of a standard C3 grass tile within JULES and C4 crops were given the values of a standard C4 grass tile. Osborne et al. (2015) speculated that an improved fit to observations could be obtained if these parameters were tuned to be more crop-specific.

The other published study using JULES-crop to date, Williams and Falloon (2015), used the global set up and the generic parameterisation of the four main crops from Osborne et al. (2015) to investigate the sensitivity of the yield from JULES-crop to the driving data variables, assessing both the relative importance of different variables and whether there is an advantage to using subdaily driving data rather than using daily driving data and performing an internal disaggregation to subdaily timescales. It also investigated the effect on the yield of initialising the model from climatology. No attempt was made to find more appropriate crop parameter values.

In this model evaluation paper, we use the observations available at the Mead FLUXNET sites US-Ne1, US-Ne2 and US-Ne3 to investigate how well each individual component of JULES performs for maize and how much of an improvement can be achieved by using more appropriate parameter values, taking into account advances in the JULES code since the Osborne et al. (2015) study. We will use these new sets of parameters in JULES-crop runs for irrigated maize at Mead.

This paper is organised as follows. Section 2 describes the JULES-crop model and the other relevant parts of the JULES code. Section 3 gives information about the observations and the model set-up used for the runs presented in this paper, both those with and without the crop model switched on. Section 4 compares the model against the observations. Section 5 contains an overall assessment about the suitability of the model for modelling maize at these sites and discusses ways that the model could be improved. It also comments on the more general applicability of the parameters and methods used in this paper for tuning JULES-crop for other sites and varieties.

## 2 Model description

In this section, we will summarise the relevant features of JULES and the JULES-crop parameterisation within it, paying particular attention to new model features available since the Osborne et al. (2015) study (i.e. post version 4.0). These new options are indicated in Table 1, Table 2, Table 3 and Table 4.





## 2.1 Crop model

In JULES-crop, the development status of each crop within a gridbox is parametrised by a crop development index (DVI). DVI is -2 before sowing, -1 at sowing, 0 at emergence and 1 at flowering. Under favourable conditions, harvest occurs at a DVI of 2. The DVI has three main functions within the JULES-crop model: it determines the harvest date, the partitioning of NPP between the crop carbon pools and the dependence of the specific leaf area on leaf carbon.

The increase in DVI over the course of the crop's lifetime is determined by crop-specific thermal time parameters, set by the user. If the dependence on photoperiod length is neglected (as in Osborne et al. (2015)), thermal time becomes an accumulation of effective temperature between one development stage and the next, where effective temperature is defined by

$$
T_{\mathrm{eff}} = \begin{cases}
0 & \text{for} \quad T < T_b \\
T - T_b & \text{for} \quad T_b \leq T \leq T_o \\
(T_o - T_b)(1 - \dfrac{T - T_o}{T_m - T_o}) & \text{for} \quad T_o < T < T_m \\
0 & \text{for} \quad T \geq T_m
\end{cases}
\tag{1}
$$

i.e. a triangular function, peaking at $T = T_o$, which is zero below $T = T_b$ and above $T = T_m$. $T_o$, $T_b$ and $T_m$ are parameters specified by the user for each crop.

Crop growth is modelled by accumulating net primary productivity over the course of a day ($\mathrm{NPP^{acc}}$) and splitting this carbon between the crop root, stem, leaf, harvest and reserve carbon pools for that tile ($C_{\mathrm{root}}, C_{\mathrm{leaf}}, C_{\mathrm{stem}}, C_{\mathrm{harv}}, C_{\mathrm{resv}}$ resp.) according to

$$
\begin{aligned}
\Delta C_{\mathrm{root}} &= p_{\mathrm{root}} \mathrm{NPP^{acc}} \\
\Delta C_{\mathrm{leaf}} &= p_{\mathrm{leaf}} \mathrm{NPP^{acc}} \\
\Delta C_{\mathrm{harv}} &= p_{\mathrm{harv}} \mathrm{NPP^{acc}} \\
\Delta C_{\mathrm{stem}} &= p_{\mathrm{stem}} \mathrm{NPP^{acc}} (1 - \tau) \\
\Delta C_{\mathrm{resv}} &= p_{\mathrm{stem}} \mathrm{NPP^{acc}} \tau,
\end{aligned}
\tag{2}
$$

where $\tau$ is the fraction of stem carbon that is partitioned into the stem reserve pool (containing the remobilizable carbohydrates) and $p_i$ (for $i = $ root, stem, leaf, harv) are the partition coefficients defined by

$$
p_i = \frac{\exp[\alpha_i + \beta_i \mathrm{DVI}]}{\sum_j \exp[\alpha_j + \beta_j \mathrm{DVI}]},
\tag{3}
$$

where $j = $ root, stem, leaf, harv. $\alpha_{\mathrm{harv}}$ and $\beta_{\mathrm{harv}}$ are both set to zero. All other $\alpha_i$ and $\beta_i$ are set by the user for each crop. Note that $\sum_j p_j = 1$.

The crop carbon pools are initialised at $\mathrm{DVI_{init}}$, which is at or just after emergence. At initialisation, the crops are given a certain amount of carbon $C_{\mathrm{init}}$, which is distributed between the carbon pools according to the values of $p_i$ at $\mathrm{DVI} = \mathrm{DVI_{init}}$.

Once $p_{\mathrm{stem}}$ drops below 0.01, carbon from the stem reserve pool is mobilised to the harvest pool, by reducing $C_{\mathrm{resv}}$ by 10% each day and adding this carbon to the harvest pool (as proposed in de Vries et al. (1989)). Similarly, once the DVI is above a





threshold value $\mathrm{DVI_{sen}}$, carbon from the leaf pool is mobilised to the harvest pool, by reducing $C_{\mathrm{leaf}}$ by a fraction

$$\mu\left(\mathrm{DVI} - \mathrm{DVI_{sen}}\right)^{\nu} \tag{4}$$

each day when $\mathrm{DVI} > \mathrm{DVI_{sen}}$ and adding this carbon to $C_{\mathrm{harv}}$, to simulate leaf senescence.

After $\mathrm{DVI_{init}}$ and if sowing date is prescribed, the model harvests the crop and resets the crop tile if any of the following conditions are satisfied:

1. DVI reaches 2 (i.e. the desired harvest condition)

2. LAI>15, since once the model reaches such large LAI it is clearly unrealistic,

3. the temperature of the second soil layer from the top falls below a user defined temperature $T_{\mathrm{mort}}$ at any time after DVI=1

4. DVI > 1.0, the carbon in the roots, leaves, stem and stem reserve pool of the crop falls below $C_{\mathrm{init}}$ and the amount of carbon in the harvest pool is greater than zero

5. the crop age reaches 1 year, so that a new crop can be sown each year.

The crop height $h$ is calculated from the $C_{\mathrm{stem}}$ pool using

$$h = \kappa \left(\frac{C_{\mathrm{stem}}}{f_{C,\mathrm{stem}}}\right)^{\lambda}, \tag{5}$$

where $\kappa$ and $\lambda$ are allometric constants and $f_{C,\mathrm{stem}}$ is the fraction of carbon in the dried stem (excluding the stem reserves), all given as input by the user.

The green (i.e. photosynthesising) leaf area index (LAI) is calculated from the leaf carbon and the specific leaf area (SLA) by

$$\mathrm{LAI} = \frac{C_{\mathrm{leaf}}}{f_{C,\mathrm{leaf}}}\mathrm{SLA}, \tag{6}$$

where $f_{C,\mathrm{leaf}}$ is the carbon fraction of the dry leaves. The SLA depends on the DVI via

$$\mathrm{SLA} = \gamma\left(\mathrm{DVI} + 0.06\right)^{\delta}, \tag{7}$$

where $\gamma$ and $\delta$ are allometric constants which are set by the user.

JULES-crop outputs water-limited potential yield if irrigation is switched off and potential yield if irrigation is on, expressed in kg C m$^{-2}$. This yield is calculated by multiplying the value of $C_{\mathrm{harv}}$ on the day of harvest by a parameter $f_{\mathrm{yield}}$ supplied by the user, which represents the fraction of $C_{\mathrm{harv}}$ that is economically valuable i.e. the maize kernel in our runs.





## 2.2 Relationship between LAI, canopy height and plant carbon for natural vegetation

When the crop model is switched off, different allometric functions are used to approximate the carbon in the leaf, stem and root pools based on the prognostics LAI and canopy height $h$. These allometric functions make use of a 'balanced' leaf area index ($\text{LAI}_{\text{bal}}$), which is calculated from canopy height using

$$\text{LAI}_{\text{bal}} = \left( \frac{a_{ws}\eta_{sl}}{a_{wl}} h \right)^{\frac{1}{b_{wl}-1}},$$
(8)

where $a_{ws}$, $a_{wl}$, $\eta_{sl}$ and $b_{wl}$ are all allometric constants, defined in relation to the respiring stem carbon $\mathcal{S}$ and the total stem carbon $\mathcal{W}$:

$$\mathcal{S} = \eta_{sl} h \text{LAI}_{\text{bal}}$$
(9)

$$\mathcal{W} = a_{ws}\mathcal{S}$$
(10)

$$\mathcal{W} = a_{wl}\left(\text{LAI}_{\text{bal}}\right)^{b_{wl}}.$$
(11)

We assume here that $\mathcal{S}$ is equivalent to $C_{\text{stem}}$ and $\mathcal{W}$ is equivalent to $C_{\text{stem}} + C_{\text{resv}}$ in the crop model. Therefore, $a_{ws}$ is equivalent to $1 - \tau$ in the crop model and these equations can be compared directly to Eq. 5 until the start of the remobilisation of the crop stem reserve pool.

The size of the leaf carbon pool $C_{\text{leaf}}$ is calculated by multiplying the LAI by the canopy-averaged specific leaf density $\sigma_l$ (in kg C (m$^2$ leaf)$^{-1}$), which is assumed to be constant i.e.

$$C_{\text{leaf}} = \sigma_l\text{LAI}.$$
(12)

The root carbon $C_{\text{root}}$ is approximated by

$$C_{\text{root}} = \sigma_l\text{LAI}_{\text{bal}}.$$
(13)

## 2.3 Canopy

JULES has a number options for calculating the photosynthetically active radiation (PAR) available to leaves at different depths in the plant canopy. In this discussion, we will focus on the canopy radiation scheme used in Osborne et al. (2015) (`can_rad_mod` 5) and the canopy radiation scheme currently recommended for layered canopies in JULES (`can_rad_mod` 6), which both treat the direct and diffuse components of the incident radiation separately (as in Sellers (1985)) and include sunflecks. We also assume a zenith angle dependence (`l_cosz`=T). JULES assumes that the incident PAR is half of the incident shortwave radiation. The amount of incident PAR composed of diffuse radiation is given as part of the driving data. The canopy is split into 10 equal layers of green leaf area index (LAI). The equations for absorption and scattering at each layer for the incident diffuse beam and the incident direct beam (including the zenith angle dependence) are solved separately, taking into account the distribution of leaf angles. The sunlit fraction of the leaf is also calculated, and absorbs light from the direct component of the direct beam radiation ("sunflecks"), in addition to the diffuse light from the direct beam and light from





the diffuse beam. The shaded fraction of the leaf absorbs light scattered from the direct beam and light from diffuse beam only (i.e. no direct sunlight). JULES has two leaf angle distributions currently implemented - spherical and horizontal. As of JULES version 4.6, JULES also includes a canopy clumping factor $a$, which scales LAI within the canopy radiation scheme and represents variation within and across canopy structures.

## 2.4 Modelling C4 photosynthesis

In JULES, potential leaf-level photosynthesis (unstressed by water availability and ozone effects) is calculated as the smoothed minimum of three rates, following Collatz et al. (1991, 1992): (a) the Rubisco-limited rate $W_c$, which depends on the maximum rate of carboxylation of Rubisco, (b) the light-limited rate $W_{\mathrm{light}}$ and (c) the rate associated with the transport of photosynthetic products for C3 plants or PEP-Carboxylase limitation for C4 plants $W_e$.

For C4 plants, $W_c$ is set to the maximum rate of carboxylation of Rubisco, $V_{\mathrm{cmax}}$. $V_{\mathrm{cmax}}$ is calculated using

$$V_{\mathrm{cmax}} = \frac{V_{\mathrm{cmax,norm}} f_T(T_c)}{\left[1 + e^{0.3(T_c - T_{\mathrm{upp}})}\right]\left[1 + e^{0.3(T_{\mathrm{low}} - T_c)}\right]} \tag{14}$$

where

$$f_T = Q_{10,\mathrm{leaf}}^{0.1(T_c - 25)} \tag{15}$$

and $T_c$ is the leaf temperature (which does not vary through the canopy in JULES) and $V_{\mathrm{cmax,norm}}$ is a normalisation constant. Note that $V_{\mathrm{cmax,norm}}$ is not $V_{\mathrm{cmax}}(T_c{=}25°C)$ but, for default JULES C3 grass and C4 grass parameters, $V_{\mathrm{cmax,norm}}$ and $V_{\mathrm{cmax}}(T_c{=}25°C)$ are within 5 % of each other. $T_{\mathrm{upp}}$ and $T_{\mathrm{low}}$ are used to give the leaf an optimum temperature range, which is superimposed on the Q10 dependence in $f_T$.

If trait-based physiology is switched off in JULES (`l_trait_phys=F`),

$$V_{\mathrm{cmax,norm}} = n_e n_l \tag{16}$$

where $n_l$ is the mass of nitrogen per mass of carbon in the leaf (with units kg N (kg C)$^{-1}$), which varies through the canopy, and $n_e$ is a normalisation constant, fitted to data. The input parameters specified by the user are $n_l^0$ ($n_l$ at the top of the canopy) and $n_e$.

In the JULES canopy radiation scheme `can_rad_mod` 5, $V_{\mathrm{cmax,norm}}$ is assumed to vary through the canopy according to $\exp(-k_n \mathrm{LAI}_{\mathrm{layer}}/\mathrm{LAI})$. In `can_rad_mod` 6, $V_{\mathrm{cmax,norm}}$ varies through the canopy according to $\exp(-k_{nl} \mathrm{LAI}_{\mathrm{layer}})$. $k_n$ and $k_{nl}$ are pft-dependent parameters set by the user.

The light-limited rate of leaf photosynthesis for C4 plants is calculated in JULES using

$$W_{\mathrm{light}} = \alpha I_{\mathrm{APAR}} \tag{17}$$

where $\alpha$ is the quantum efficiency in mol $CO_2$ (mol PAR photons)$^{-1}$ and $I_{\mathrm{APAR}}$ is the absorbed photosynthetically active radiation (APAR) in mol PAR photons m$^{-2}$ s$^{-1}$. As discussed, `can_rad_mod` 5 and `can_rad_mod` 6 include the effect of

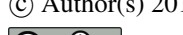

sunflecks by spitting the leaf into a sunlight and a shaded part, which have different values of $I_{\mathrm{APAR}}$ and therefore different $W_{light}$.

The rate associated with PEP-Carboxylase limitation $W_e$ in JULES is

$$W_e = 2 \times 10^4 V_{\mathrm{cmax}} \frac{c_i}{P_*} \tag{18}$$

where $P_*$ is the surface air pressure and $c_i$ is the leaf internal carbon dioxide partial pressure, which is calculated for C4 plants using

$$c_i = (c_a - \Gamma) f_0 \left( 1 - \frac{\Delta q}{\Delta q_{\mathrm{crit}}} \right) + \Gamma \tag{19}$$

where $\Gamma$ is the photorespiration point (zero for C4 plants) and $c_a$ is canopy $CO_2$ pressure. $\Delta q$ is the canopy level specific humidity deficit, $\Delta q_{\mathrm{crit}}$ is the critical specific humidity deficit and $f_0$ is the ratio of $c_i$ to $c_a$ at which the canopy level specific humidity deficit is zero. $c_a$ is calculated from $R_{CO_2} P_* / \epsilon$, where $R_{CO_2}$ is the atmospheric $CO_2$ mass mixing ratio and $\epsilon = 1.5194$ is the ratio of molecular weights of $CO_2$ and dry air. As an example, for zero specific humidity deficit, an atmospheric $CO_2$ mass mixing ratio of $5.6 \times 10^{-4}$ (2003 global average, Dlugokencky and Tans (2016)), $f_0 = 0.8$ (JULES C4 grass default), the value of $W_e$ is $5.9 V_{\mathrm{cmax}}$.

The rate of gross leaf photosynthesis $W$ is the smoothed minimum of $W_c$, $W_{\mathrm{light}}$ and $W_e$ (calculated using non-rectangular hyperbolic functions with the curvature parameters hard-wired). The net potential (i.e. unstressed) leaf photosynthetic carbon uptake $A_p$ is the gross leaf photosynthesis minus the dark leaf respiration $R_d$. The potential leaf photosynthesis is converted to a net photosynthesis by multiplying by a soil water stress parameter $\beta$. Stomata at points with negative or zero net photosynthesis or where the leaf resistance exceeds its maximum value are closed (i.e. leaf gross photosynthesis is zero). Leaf resistance is calculated from the net (i.e. water-limited) rate of photosynthesis, $(c_a - c_i)$, the leaf temperature and the ratio of leaf resistance for $CO_2$ to leaf resistance for $H_2O$ (=1.6).

## 2.5 Respiration

In JULES, the (non-water limited) leaf dark respiration $R_d$ (in mol $CO_2$ $(m^2$ leaf$)^{-1}$ $s^{-1}$) is calculated by

$$R_d = \begin{cases} 0.7 f_{dr} V_{\mathrm{cmax}} & \text{for} \quad I_{\mathrm{APAR}} \Delta \mathrm{LAI} > 10 \mu \mathrm{mol}\, CO_2 (m^2 \mathrm{ground})^{-1} s^{-1} \\ f_{dr} V_{\mathrm{cmax}} & \text{otherwise} \end{cases} \tag{20}$$

to allow for the inhibition of dark respiration during daylight. $R_d$ is summed over the canopy levels for sunlit and shaded leaves to get $R_{dc}$, the canopy dark respiration in (in mol $CO_2$ $(m^2$ ground$)^{-1}$ $s^{-1}$).

The plant maintenance respiration in kg C $(m^2$ ground$)^{-1}$ $s^{-1}$ is calculated (for the setting `l_scale_resp_pm=T`) using

$$
\begin{aligned}
R_{pm} &= 0.012 R_{dc} \beta \left( 1 + \frac{N_{\mathrm{root}}}{N_{\mathrm{leaf}}} + \frac{N_{\mathrm{stem}}}{N_{\mathrm{leaf}}} \right) \tag{21} \\
&= 0.012 R_{dc} \beta \left( 1 + \mu_{rl} \frac{C_{\mathrm{root}}}{C_{\mathrm{leaf}}} + \mu_{sl} \frac{C_{\mathrm{stem}}}{C_{\mathrm{leaf}}} \right), \tag{22}
\end{aligned}
$$





where $N_{\text{root}}, N_{\text{stem}}, N_{\text{leaf}}$ are the nitrogen in the roots, stems and leaves respectively. $\mu_{rl}$ is the mass ratio of nitrogen to carbon in the roots divided by the ratio of nitrogen to carbon in the leaves. $\mu_{sl}$ is the mass ratio of nitrogen to carbon in the stem (not including stem reserves) divided by the ratio of nitrogen to carbon in the leaves. The factor 0.012 relates mol $CO_2$ to kg C. If the option `l_scale_resp_pm=F` is set, the root and stem terms do not depend on $\beta$.

In JULES, plant growth respiration $R_{pg}$ is a fixed fraction $r_g$ (the growth respiration coefficient) of the gross primary productivity ($\Pi_G$) minus the plant maintenance respiration:

$$R_{pg} = r_g(\Pi_G - R_{pm}). \tag{23}$$

Note that this relation results in the correct growth respiration on timescales of order of a day or longer (on the model timestep scale, $R_{pg}$ will be negative in the night, which is misleading if taken in isolation). The net primary productivity $\Pi_N$ is therefore

$$\begin{aligned} \Pi_N &=& \Pi_G - R_{pm} - R_{pg} & \tag{24} \\ &=& (1 - r_g)(\Pi_G - R_{pm}). & \tag{25} \end{aligned}$$

## 2.6 Irrigation

In JULES, irrigation is implemented such that the water in the top two soil layers is continuously topped up to a critical level (often the field capacity) during the 'irrigation season', if sufficient irrigation water is available. We will consider the irrigation
season to last all year (`irr_crop=0`) and treat the supply of irrigation as unlimited (`l_irrig_limit=F`). With these settings, the soil water stress parameter $\beta$ stays approximately equal to one i.e. the plant is not water-stressed.

When irrigation is on, the root distribution has a negligible influence on model performance.

## 2.7 Nitrogen limitation

Although JULES has a nitrogen cycle implemented (as of version 4.4), it can not yet be used in conjunction with the crop
model. We therefore make the assumption here that the crops are not nitrogen limited.

## 3 Experimental Set-up

### 3.1 Observations

There are three FLUXNET sites at the University of Nebraska Agricultural Research and Development Center near Mead, Nebraska, which are located within 1.6 km of each other: US-Ne1, US-Ne2 and US-Ne3. Both US-Ne1 and US-Ne2 are
irrigated with a central pivot system, whereas US-Ne3 is entirely rainfed (Verma et al., 2005; Suyker et al., 2004, 2005). US-Ne1 grows maize, whereas US-Ne2 and US-Ne3 are maize-soybean rotations. The observations span 2001 to 2015 (although not all variables were available for this entire period).

The observations of the biomass of green leaves, yellow leaves, stem and reproductive parts of maize (kernel, cob, husk, ear shank, silk) were made after the plant material was dried to a constant temperature of 105 degrees Celsius. In the observations,




green leaves encompasses all green leaf material from the collar to the leaf tip, yellow leaves are defined as greater than 50% necrotic (or entirely yellow) leaf and the stem includes stem, leaf sheaths, immature or undeveloped ears and unfurled leaves.

Hourly incident and absorbed PAR (400 to 700 nm) observations are available from the Mead FLUXNET site. Absorbed PAR was calculated using two point quantum sensors above the canopy, pointing up and down, and two line quantum sensors below the canopy, pointing up and down. The line quantum sensors below the canopy integrate over an area 1cm by 1m, in order to even out effects such as sunflecks.

The observations were used in three ways: to determine the input parameters to the JULES runs (air temperature, carbon pools, leaf nitrogen, absorbed PAR, canopy height, LAI), to drive the JULES runs themselves (meteorological variables, LAI, canopy height) and to compare the JULES run results against (GPP, carbon pools, LAI, canopy height).

## 3.2 Model setup

Two types of JULES runs were used in this study:

1. Maize is treated as a natural PFT tile (i.e. crop model is switched off), with LAI and crop height prescribed from observations (linearly interpolated to create a daily time series).

2. Maize is considered as a crop tile (i.e. crop model is switched on).

The runs were driven by hourly observations of downward shortwave radiation, downward longwave radiation, precipitation, air temperature, wind speed, pressure, specific humidity and diffuse radiation fraction. Each year and site was modelled as a separate run, each starting on the 1st March. Annual global $CO_2$ atmospheric concentrations were taken from Dlugokencky and Tans (2016).

A summary of the model input parameters used in both types of runs are given in Table 1, Table 2, Table 3, Table 4. The following sections describe in more detail how the choice of input parameters was made. In setting these parameters, it was assumed that there was no limitation from nitrogen availability or water availability.

### 3.3 Parameters required for crop tiles only

### 3.3.1 Crop development parameters

The cardinal temperatures $T_b$, $T_o$, $T_m$ in this analysis have been kept the same as Osborne et al. (2015), which were chosen based on the literature review in Sánchez et al. (2014). As in Osborne et al. (2015), there was assumed to be no dependence of thermal time on the photoperiod.

The thermal times were calculated using the available Mead data for the sowing date, the date at which 50% of the plants had emerged[1], the date at which 50% of the plants were at the R1 or 'estimated R1' growth stage (i.e. had begun the reproductive phase), the date at which 50% of the plants had reached the R6 growth stage (maturity) and the harvest date, together with the observed hourly air temperature and Eq. 1. These thermal times are given in Table 5. In the runs presented in Section 4,

---

[1]emergence dates for 2001-3 were estimated by the site investigator based on weather.





the thermal times for sowing to emergence, emergence to flowering and flowering to harvest for each year at a site are used in JULES-crop directly, to simulate the crop development as closely as possible for a finished crop season, where the harvest date is known.[2]

The sowing date is prescribed (i.e. `l_prescsow=T`). An option for sowing date to be calculated dynamically using rate of change of day length and soil temperature and moisture does exist (`l_prescsow=F`), but this is not considered here as it is still under development and not recommended for use (Osborne et al., 2015).

Since harvest dates are available, $T_{\mathrm{mort}}$ was set low enough that it did not trigger harvest.

---

[2]Since these thermal times are meant to represent intrinsic properties of the cultivar, it would be interesting to investigate the use of the mean thermal times. However, unlike the date of physiological maturity, the harvest date depends more practical management conditions. In situations where modelling the yield is more important than modelling the time series of the fluxes, for example, it might be more appropriate to recalibrate the DVI such that the crop reaches DVI=2 at physiological maturity and is harvested immediately.



### 3.3.2 Carbon partitioning

The carbon partitioning parameters $\alpha_i, \beta_i$ were tuned to observations of the biomass of green leaves, yellow leaves, stem and reproductive parts of maize. The ratio of carbon to biomass in each part of the plant was assumed to be the same and constant in time. The $C_{\mathrm{leaf}}$ pool in the model contains green leaves only (since $C_{\mathrm{leaf}}$ is directly linked to LAI and photosynthesis) and the

5 $C_{\mathrm{harv}}$ pool consists of both the reproductive parts of the plants and the yellow leaves. Stem carbon in the model is split between the $C_{\mathrm{stem}}$ and $C_{\mathrm{resv}}$ pools. The biomass observations were linearly interpolated to get a daily time series and then differentiated with respect to time. Ratios of these rates were then plotted as a function of DVI (Figure 1). Using these plots alongside the function for root carbon from de Vries et al. (1989) (since there were no direct measurements of root biomass available from the Mead sites), new, tuned values for $\alpha_i, \beta_i$ were found. These tuned parameters (dashed lines) show an improvement in the

10 proportion of the increase in above-ground carbon that goes to the green leaves (Figure 1, top) and the proportion of the increase in above-ground carbon that goes to the stem (Figure 1, middle) for DVI < 0.8 as compared to the parameters used in Osborne et al. (2015) (solid line). However, note that, even after the tuning, the proportion of the increase in above-ground carbon that goes to the green leaves does not drop off sharply enough for DVI > 0.8 compared to the observations. The tuned partition fractions are shown more clearly in Figure 2 (colours), together with the functions given in de Vries et al. (1989) (the $\alpha_i, \beta_i$ in

Osborne et al. (2015) were fitted to these functions with minor adjustments as a result of global runs). It was not possible to fit $p_{\mathrm{root}}$ accurately to the expression from de Vries et al. (1989) for approximately DVI 1.0 to 1.4 given the constraints above. In addition, in reality, water stress can also increase the fraction of NPP going to the roots (see discussion in e.g. de Vries et al. (1989) and Song et al. (2013)), but this effect is not taken into account in JULES-crop.

### 3.3.3 Remobilisation of stem carbon

The stem biomass observations were used to tune the value for the stem reserve remobilisation constant $\tau$. The relation governing the stem reserve remobilisation can be rearranged to

$$1 - \frac{M_{\mathrm{stem}}}{M_{\mathrm{stem}}^{\mathrm{max}}} = \tau \left( 1 - 0.9^{d_{\mathrm{max}}} \right), \tag{26}$$

where $M_{\mathrm{stem}}$ is stem biomass (including reserves), $M_{\mathrm{stem}}^{\mathrm{max}}$ is the maximum value of $M_{\mathrm{stem}}$ in that site in that year and $d_{\mathrm{max}}$ is the day since $M_{\mathrm{stem}}^{\mathrm{max}}$ occurred.

Therefore, plotting $1 - \frac{M_{\mathrm{stem}}}{M_{\mathrm{stem}}^{\mathrm{max}}}$ against $\left( 1 - 0.9^{d_{\mathrm{max}}} \right)$ should give a straight line with gradient $\tau$. Using the assumption that the day with maximum stem biomass was approximately the same day as the day with the maximum stem biomass measurement, a straight line was fitted to the observations and an approximate value of $\tau = 0.12$ was obtained. However, as can be seen in Figure 3 (which displays both the new, tuned value $\tau = 0.12$ (black, dashed line) and the value used in Osborne et al. (2015) of $\tau = 0.35$ (black, solid line), which was obtained from de Vries et al. (1989)), this parameterisation does not capture the large

spread in the observations (blue, green and red lines). The uncertainty this introduces into the model is not critical, since there are no strong feedbacks involved (unlike, for example, uncertainty in SLA just after emergence), but it will affect the outputted yield.





### 3.3.4 Senescence

The observations of green leaf biomass and above-ground biomass were used to tune the senescence parameters $\mu$, $\nu$ and $\mathrm{DVI_{sen}}$. The above-ground biomass measurements were combined with the partition fractions from Section 3.3.2 and the carbon to biomass ratios from Section 3.3.7 and the senescence parametrisation from Eq. 4 to get a time series for green leaf biomass (Figure 4, black lines). This could then be compared to the observed time series for green leaf biomass (Figure 4, coloured lines). It is clear that, if the parameterisation from Osborne et al. (2015) is used (Figure 4, left plot, solid black lines), senescence starts late and then progresses too abruptly as compared to the observations. However, with the new parameterisation, with the new free parameters $\mu$, $\nu$ and $\mathrm{DVI_{sen}}$, it is possible to get a much better fit to the observations (Figure 4, right plot, dashed black lines). Note that this tuning partially compensates for the bias in the proportion of carbon going to the leaves between DVI 0.8 and 1.0 in Figure 1 (top). If this bias was not present, senescence could start more gradually, which would enable a better fit to leaf carbon around DVI=1.75. Also, the tuned lines underestimate the leaf biomass at around DVI=1.75, which will help to compensate for the model being unable to capture the drop in photosynthetic capacity in the green maize leaves towards the end of the season.

### 3.3.5 Crop height

Stem biomass measurements up until the maximum in each year and the corresponding crop height measurements from the Mead FLUXNET sites were used to fit the allometric constants $\kappa$ and $\lambda$, through rearranging Eq. 5 to $h = \kappa' M_{\mathrm{stem}}^{\lambda}$ where $\kappa' = \kappa (1 - \tau)^{\lambda}$. For consistency, it is important that the $\tau$ used in this expression is the same value as the $\tau$ used in Eq. 2. Figure 5 shows the observations (points), along with the fit using parameters from Osborne et al. (2015) (solid black line, $\lambda = 0.4$, $\kappa' = 3.06$) and a tuned fit (dashed black line, $\lambda = 0.38$, $\kappa' = 3.43$).

### 3.3.6 Specific leaf area

The allometric constants $\gamma$ and $\delta$ relating specific leaf area to DVI (Eq. 7) are tuned using Figure 6, which plots SLA observations against DVI (points), the tuned fit (dashed line) and the parameters used in Osborne et al. (2015) (solid line). The crop in the model is very sensitive to SLA for low values of DVI because of the feedback between leaf area index and leaf carbon. The model lines in Figure 6 have the steepest gradient for low values of DVI, where there is also a greater spread of observations.

### 3.3.7 Carbon to biomass ratio in stem and leaves

The observations of carbon fraction of the green leaf biomass (canopy mean) against day after sowing is shown in Figure 7. The mean of all of these observation together is 0.43, although there are possible indications of a slight downward trend in each site in each year with time, which would indicate that this value might be sensitive to the dates on which the carbon is measured and all three sites in 2001 have lower values. In these runs we have used $f_{\mathrm{C,leaf}} = f_{\mathrm{C,stem}}$=0.439. For comparison, de Vries et al. (1989) gives the carbon fraction of leaves and stems for non-leguminous and no-rice crops as 0.459 and 0.494 respectively and Osborne et al. (2015) used $f_{\mathrm{C,leaf}} = f_{\mathrm{C,stem}}$=0.5.



### 3.3.8 Initial amount of carbon in crops

Assuming that, near emergence, approximately half of the plant carbon is above ground (Figure 2), values for the parameters governing initialisation of $C_{\mathrm{init}} = 8.0 \times 10^{-4}$ and $\mathrm{DVI}_{\mathrm{init}} = 0.1$ can be derived from the above-ground biomass measurements plotted in Figure 8 and the carbon to biomass ratios. Since there are no measurements below DVI=0.1, and the model is very sensitive to these parameters, we do not attempt to set a $\mathrm{DVI}_{\mathrm{init}}$ below 0.1 and extrapolate. Note also that the initial value of carbon is very sensitive to the thermal time for emergence. Theoretically, the initial amount of carbon should also be proportional to the planting density, although this difference is not apparent in Figure 8. Figure 8 also shows that the value $C_{\mathrm{init}} = 1.0 \times 10^{-2}$ which was used in Osborne et al. (2015) to initialise the crop at $\mathrm{DVI}_{\mathrm{init}} = 0.0$ is too high to be consistent with the above-ground biomass observations.

### 3.3.9 Yield fraction

As discussed above, the $C_{\mathrm{harv}}$ pool in JULES contains both the reproductive parts of the maize crop (kernel, cob, husk, ear shank and silk) and the yellow leaf carbon and the proportion of this carbon pool that contributed to yield carbon $f_{\mathrm{yield}}$ is set by the user. The value of $f_{\mathrm{yield}}$ can be derived using the latest observations in each season of the biomass of the reproductive part of the crop, the proportion of this reproductive biomass which is composed of kernels, and the yellow leaf biomass. The yield fraction is then calculated as the kernel fraction of the sum of the reproductive part of the crop and the yellow leaves, leading to an approximate value of $f_{\mathrm{yield}} = 0.74$ (Figure 9). This assumes that there is no significant change in $f_{\mathrm{yield}}$ between the last measurement of the season and the harvest and also that the carbon fraction of the biomass in the both the reproductive parts and the yellow leaves is the same. Typically, an accurate value of $f_{\mathrm{yield}}$ is not important in impact studies, since this constant can be incorporated into a yield gap parameter.

### 3.4 Parameters required by natural pft tiles only

To obtain the allometric parameters required to relate the plant carbon pools to plant height and LAI when the crop model is switched off, $\mathrm{LAI}_{\mathrm{bal}}$ was assumed to be approximately equal to LAI up to the maximum LAI at the site for each year. As discussed in Section 2.2, $a_{ws}$ is assumed to be equivalent to $1$-$\tau$ i.e. $a_{ws}$=0.88. The stem biomass observations can be used to obtain values for $a_{wl}$, $b_{wl}$ and $\eta_{sl}$, for a set ratio of carbon to biomass in the stem (see Section 3.3.7). First, a value for $\eta_{sl}$ of 0.017 kg C m$^{-1}$ (m$^2$ leaf)$^{-1}$ was obtained by plotting the stem biomass observations against LAI multiplied by crop height for points up until the maximum LAI for each site in a particular year (Figure 10, left). Secondly, $a_{wl}$ and $b_{wl}$ simultaneously fitted to (a) the stem biomass observations against LAI for points up until the maximum LAI for each site in a particular year (Figure 10, right), (b) crop height against stem biomass observations, up until the maximum stem biomass measurement for each site in a particular year (Figure 11) and (c) $\mathrm{LAI}_{\mathrm{bal}}$ against LAI up until the maximum LAI for each site in a particular year (Figure 12). This gave $a_{wl}$=9.5$\times$10$^{-3}$ kg C m$^{-2}$ and $b_{wl}$=1.767.

As we saw in Figure 6, Eq. 12 is not a good approximation for maize, particularly when DVI is less than 0.5. For the purposes of these runs, an approximate value at DVI=1 was used.





### 3.5 Parameters required by both crop tiles and natural pft tiles

#### 3.5.1 Canopy radiation scheme

The JULES default C4 grass settings for the PAR leaf scattering coefficient $\omega_{\mathrm{PAR}} = 0.17$ and the PAR leaf reflection coefficient $\alpha_{\mathrm{refl,PAR}} = 0.1$ were used (these are very similar to the values quoted in Sellers (1985) for live maize leaves: $\omega_{\mathrm{PAR}} = 0.175$,

$\alpha_{\mathrm{refl,PAR}} = 0.105$) and a spherical angle distribution. These are the same parameter values and options that were used in Osborne et al. (2015) to model maize. The soil albedo was set to 0.133, which was the value from the nearest gridbox in the ancillary used in the HadGEM2-ES model (Collins et al., 2011; Jones et al., 2011), which was used in the Osborne et al. (2015) global runs.

    The canopy clumping factor was tuned by comparing the fraction of incident PAR absorbed by the canopy (FAPAR), us-

ing absorbed and incident PAR observations and interpolated LAI observations, to the model FAPAR, using observed diffuse radiation fraction and interpolated LAI observations up until flowering. The python package `pySellersTwoStream` (downloaded 15.09.2016) was used to calculate the model FAPAR since it is able to reproduce the results of the JULES radiation scheme exactly but can be called directly from our (python) analysis scripts, without the need for extra JULES runs for each combination of parameters tested.

Absorbed PAR through the canopy in the model closely follows a exponential decay function. Calculating FAPAR involves integrating this exponential decay over the canopy: Figure 13 (centre row) shows the resulting FAPAR distribution against total LAI for a uniform canopy (canopy clumping factor $a = 1$). For mostly direct radiation (diffuse radiation fraction 0.2-0.3), the rate of decay with layer LAI in the model shows a clear dependence on the zenith angle (Figure 13, centre right), whereas for mostly diffuse radiation (diffuse radiation fraction 0.8-0.9), this zenith angle dependence is greatly reduced (Figure 13,

centre left). While the observations (Figure 13, top row) also show a strong zenith angle dependence as the fraction of diffuse radiation decreases, the observations are, in general, consistent with a much lower effective decay constant (in particular, the model FAPAR values are higher than the observations at intermediate LAI values $\sim 2$). The observed FAPAR values also have a much larger scatter than seen in the model FAPAR.

    Decreasing the canopy clumping factor is equivalent to decreasing the effective decay constant in the model. Figure 14

shows the value of the clumping factor that would be needed to reproduce each FAPAR observation, given the observed LAI and diffuse radiation fraction. While there is a large spread in clumping values derived in this way, these results appear to indicate that a clumping factor between 0.5 and 0.8 would be consistent with the majority of the observations. In this study, we therefore set $a = 0.65$. Figure 13 (bottom row) shows that using this clumping factor value to calculate model FAPAR gives a better fit to the observations, particularly for the intermediate LAI values.

Erectile, vertical and horizontal leaf angle distributions (for a uniform canopy) were also investigated, but the spherical distribution gave the best fit to the FAPAR observations.

    The FAPAR observations can not be used to tune the model once green leaf area index has started to drop significantly, as the observations include PAR absorbed by any part of the plant, whereas the JULES canopy scheme models the PAR absorbed by photosynthesising leaves only. Whether the model canopy scheme needs to be extended to include the shading of green leaves



by yellow leaves and other non-root biomass depends on the distribution of the remaining green leaves through the canopy (essentially, the model is roughly assuming that all the green LAI is at the top of the plant and so does not get shaded by other plant material). Different approaches have been used in the literature. For example, Sellers (1985) models maize assuming that green and dead leaves are evenly distributed throughout the canopy, whereas de Vries et al. (1989) says that "maximum leaf

photosynthesis in a senescencing crop declines with time. The oldest leaves in the base of the canopy are affected first".

### 3.5.2   Photosynthesis light response curve

In the literature, the photosynthetic capacity of maize leaves (per leaf area) declines with age and the older leaves are lower in the canopy Dwyer and Stewart (1986); Stirling et al. (1994). As discussed in Section 2.4, change in photosynthetic capacity through the canopy can be modelled in JULES by a non-zero $k_{nl}$, which we assume is due to change in nitrogen per unit leaf

area through the canopy.

The nitrogen per unit leaf area as a function of layer LAI at anthesis (60 days after sowing) in Massignam et al. (2001) for the highest nitrogen availability level (150 kg N ha$^{-1}$, residual soil nitrate 31 kg ha$^{-1}$) was consistent with a $k_{nl}$ of approximately 0.07. Since this is low, in this study, the variation of nitrogen per unit leaf area through the canopy is neglected i.e. $k_{nl} = 0.0$. The inclusion of a non-zero $k_{nl}$ would have the effect of increasing GPP, as the plant would be able to make more efficient use

of the incoming radiation.

In this study, trait-based physiology was switched off (i.e. `l_trait_phys` = F). However, the same results could be obtained by switching trait-based physiology on and choosing values for the new parameters that are equivalent to the ones used here.

Figure 15 shows the observations of the nitrogen mass per unit carbon mass (left) and per unit leaf area (right) averaged over

the canopy. In both plots, nitrogen rapidly decreases with time at the beginning and end of the season, which cannot be captured by JULES. The inclusion of a non-zero $k_{nl}$ would also not solve this problem, as this would simply increase the nitrogen per leaf area mid-season, as can be seen in Figure 16 for $k_{nl} = 0.2$.

In this study, the temperature dependence of $V_{cmax}$ is fixed by fitting Eq. 14 to the expression given in de Vries et al. (1989) (Figure 17). The default JULES C4 grass parametrisation of $V_{cmax}$ is more sharply peaked, has its maximum at a

higher temperature and is more asymmetrical. Also plotted is the expression for the temperature dependence for maize $V_{cmax}$ from Massad et al. (2007). Puntel (2012) modelled $V_{cmax}$ for maize at the Mead site and fit the results with MaizeGro, using the default temperature dependence, which gives a peak at approximately 33 °C . Puntel (2012) verified this relation by successfully fitting the model to results from modern maize cultivars from Kim et al. (2007), Crafts-Brandner and Salvucci (2002) and Naidu et al. (2003), which all show the peak in $V_{cmax}$ at approximately the same temperature. Puntel (2012) related

the normalisation of $V_{cmax}$ to the leaf nitrogen per biomass, for example, at 30 g N kg$^{-1}$ at the V14 growth stage, maximum assimilation at 25 °C was 37 $\mu$mol m$^{-2}$ s$^{-1}$. The temperature dependence of maize at high temperatures was looked at in more detail in Crafts-Brandner and Salvucci (2002), which included an investigation into the dependence on the rate of temperature change. The experiment with the more gradual temperature change in Crafts-Brandner and Salvucci (2002) corresponds well to the high temperature dependence of the de Vries et al. (1989) expression.





The canopy average $V_{cmax,norm}$ was tuned using the value of $V_{cmax}$ at 25 °C at 340 vppm $CO_2$ at a specific leaf weight of 450 kg h$^{-1}$, which is the canopy average at DVI=1 (for maize cv Pioneer) from de Vries et al. (1989). $n_{l0}$ was set to the approximate value of the observations in Figure 15(left) at DVI=1, which then constrains $n_e$ (since $n_e = V_{cmax,norm}/n_{l0}$ when $k_{nl} = 0$). The quantum efficiency $\alpha$ was set to the value from de Vries et al. (1989) of 0.055 $\mu$mol C m$^{-2}$ s$^{-1}$ ($\mu$mol photons m$^{-2}$ s$^{-1}$)$^{-1}$ for maize, which was quoted for temperatures lower that 45 °C (above this temperature, it drops sharply - an effect which is not reproduced in JULES). This is consistent with values in the literature (e.g. Massad et al. (2007) and references therein) and consistent with the fitted values of $\alpha$ from Puntel (2012). The value of $\alpha$ for maize is not dependent on leaf age or position (Dwyer and Stewart, 1986). This method of tuning the JULES parameters has assumed that the two limiting rates are predominantly $W_c$ and $W_{light}$, not $W_e$.

Note, however, that the photosynthesis light response curve in de Vries et al. (1989) has an exponential dependence on the absorbed radiation, which causes the shape to vary slightly from the non-rectangular hyperbolae used in JULES (with hard-wired values of curvature from Collatz et al. (1992)), leading to lower values of photosynthesis below approximately 1500 $\mu$mol photons m$^{-2}$ s$^{-1}$.

The parameters involved in calculating the leaf internal carbon dioxide partial pressure, $\Delta q_{crit}$ and $f_0$ (in Eq. 19), were not expected to strongly limit the results since this current study focusses on carbon fluxes rather than water fluxes, the runs are irrigated and the rate $W_e$ is not expected to be limiting. $\Delta q_{crit}$ was left at its default C4 grass value (as in Osborne et al. (2015)) and $f_0$ was set to 0.4 (consistent with the range of maize measurements quoted in de Vries et al. (1989)).

### 3.5.3 Respiration

Values for $\mu_{rl}$ and $\mu_{sl}$ (from Eq. 22) were obtained for maize from de Vries et al. (1989) of $\mu_{rl} = 0.39$ and $\mu_{sl} = 0.43$ (note that this assumes one constant value for the nitrogen per carbon in leaves over the crop season and $\tau = 0.12$).

Fixing the value for the dark respiration coefficient $f_{dr}$ (used in Eq. 20) is complicated by the inclusion in the code of inhibition of leaf respiration in the light. Also, Atkin et al. (1997) demonstrates that the dark respiration in darkness decreases as the time the leaf has been in darkness increases. This complicates the use of the light response curves for fitting this parameter, since this means that parameters measured during the day will not necessarily correspond to those needed in JULES for modelling the average dark respiration over a 24 hour period. Using de Vries et al. (1989) values for the maximum rate of leaf photosynthesis at 450 kg biomass per hectare and maintenance respiration at 25°C for maize gives $f_{dr}^{24h} = 0.0081$ over the course of 24 hours. Even with a correction for inhibition of dark respiration in the light, this is inconsistent with the spread of fitted values of dark respiration to maximum assimilation to light response curves measured at the site between 10:00 and 14:00 local time, presented in Puntel (2012) (leaf is exposed to ambient light pre-measurements), which are much higher, unless the dark respiration derived from the light curves is assumed to have a contribution from what JULES considers the 'growth respiration'. In general, the dark respiration coefficient estimated from light response curves for maize appears to be higher than the value derived from the maintenance respiration measurement in de Vries et al. (1989) (e.g. Collatz et al. (1992), Dohleman and Long (2009)), which is consistent with there being a component from growth respiration. In our JULES runs, we will use $f_{dr}$ derived from the maintenance respiration observation in de Vries et al. (1989), corrected assuming that in the





day of measurement 50% of leaves experienced inhibition of the dark respiration by light i.e. $f_{dr}$ is set to 0.0081/0.85=0.0095 (this assumption was later tested, and found to be accurate to within 2%).

de Vries et al. (1989) gives a growth respiration coefficient of 0.22, 0.18, 0.19, 0.18 for maize leaves, stem, roots and cob/grain respectively. These values can not be used directly in JULES since, as described earlier, the growth respiration coefficient in JULES is a constant for each carbon pool. Here, we set $r_g$ is set to 0.25 for every PFT, as in the JULES Global Land (GL4.0) configuration (Walters et al., 2013) (note however that this approximation of a constant $r_g$ for each plant carbon pool would break down for other crops e.g. soybean).

It is also worth noting that Puntel (2012) found that the maximum assimilation rate had a much stronger relationship with leaf nitrogen than the leaf dark respiration rate. In addition, Stirling et al. (1994) shows a strong dependence in dark respiration in maize over time (using fits to light response curves), which can not be captured in JULES: at degree day 220 (roughly where the leaf area reaches a maximum), it is approximately twice as high at degree day 50. As we have discussed, maintenance respiration and $V_{cmax}$ co-vary in JULES, but the growth respiration is linked to net primary productivity, which increases in the crop up until approximately anthesis. Therefore, the total leaf respiration in the model will vary in time, and will have a different dependence on time to $V_{cmax}$. However, the issues we have already identified with the modelling of the evolution of $V_{cmax}$ over time will impact the accuracy of the modelling of the maintenance component of the leaf respiration over time. Leaf dark respiration rates also differs between different maize hybrids (Earl and Tollenaar, 1998). There is therefore a large uncertainty in the parameter $f_{dr}$ and the overall determination of growth respiration.

## 4 Results

In this section we present the results from the JULES runs and compare with observations from the Mead sites. The runs with the crop model switched off and prescribed LAI and height are useful for evaluating the parameter choices for photosynthesis and respiration, without the additional complication of the feedback between LAI and NPP, so will be discussed first. The results from the full crop model configuration will then be evaluated.

### 4.1 Results from JULES runs without the crop model

### 4.1.1 Gross primary productivity

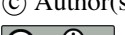



**Figure 1.** Top: ratio of rate of change of $C_{\text{leaf}}$ to rate of change of above-ground carbon $C_{ag}$, middle: ratio of rate of change of $C_{\text{stem}} + C_{\text{resv}}$ to rate of change of above-ground carbon, bottom: ratio of rate of change of $C_{\text{harv}}$ to rate of change of above-ground carbon. Solid black line uses the original crop parameters from Osborne et al. (2015), dashed black line uses the tuned parameters. Blue, green, red lines are derived from US-Ne1, US-Ne2 and US-Ne3 observations respectively.

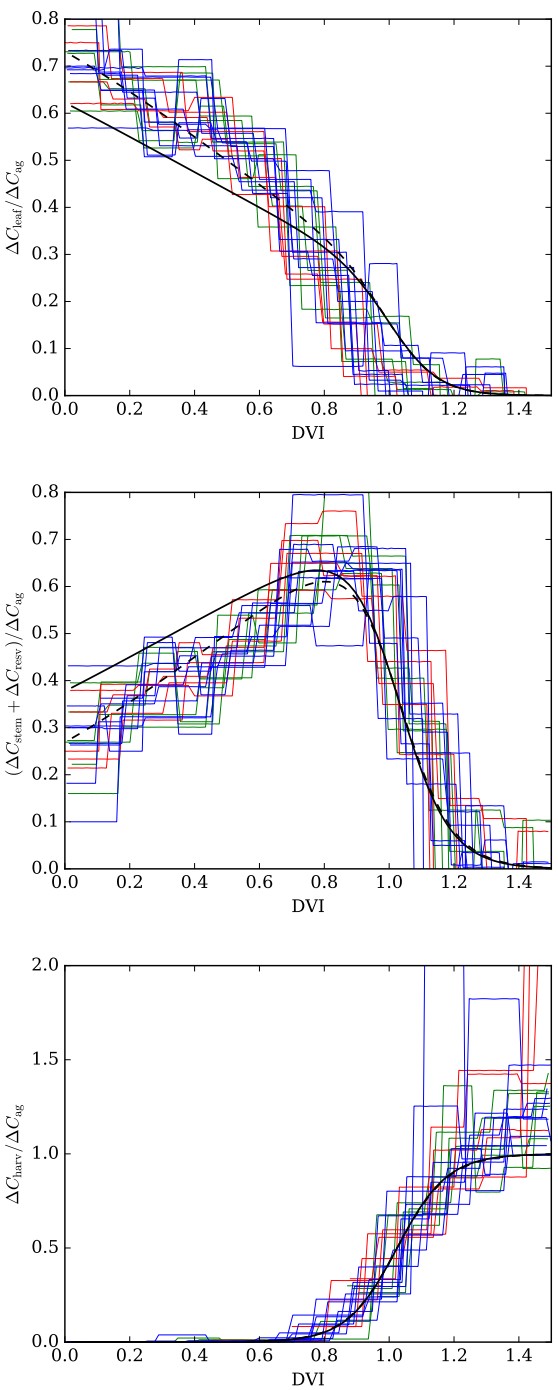





**Figure 2.** Partition fractions as a function of DVI using the tuned parameters. The dotted lines are from de Vries et al. (1989).

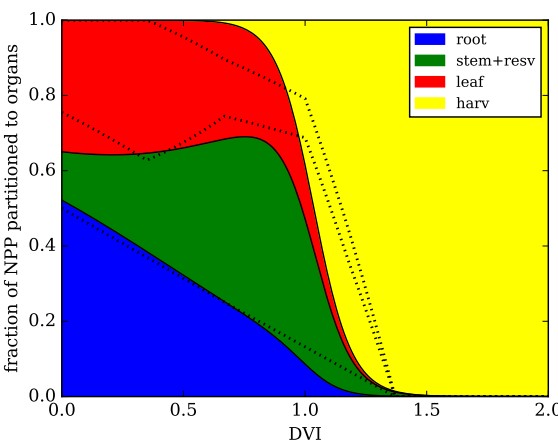

**Figure 3.** Stem biomass measurements ($M_{\text{stem}}$) normalised to the maximum measurement for that site in that year ($M_{\text{stem}}^{\text{max}}$) against day since the maximum measurement ($d_{\text{max}}$). Blue, green, red lines are derived from US-Ne1, US-Ne2 and US-Ne3 observations respectively. The dashed black line uses the tuned value $\tau = 0.12$, whereas the solid black line uses the Osborne et al. (2015) value $\tau = 0.35$.

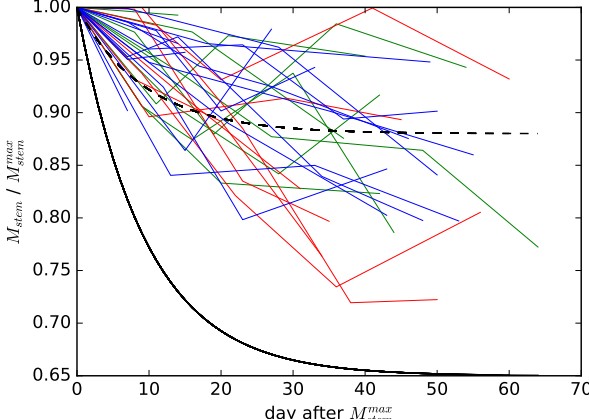





**Figure 4.** Green leaf biomass against DVI. Blue, green, red lines are derived from US-Ne1, US-Ne2 and US-Ne3 observations respectively. Black lines are generated using model parameters from Osborne et al. (2015) (left plot, solid lines) and the new, tuned parameters (right plot, dashed lines).

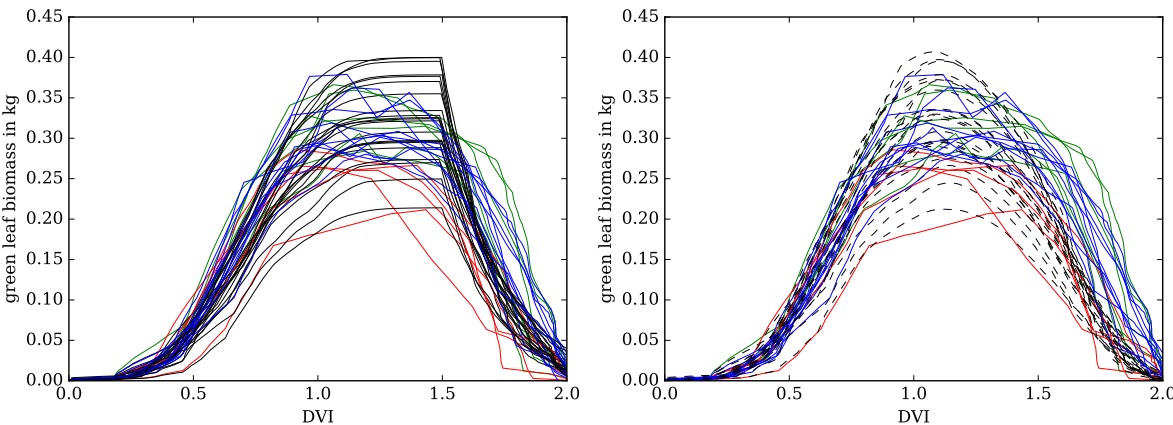

**Figure 5.** Crop height against dry stem biomass (including reserves). Crop height against dry stem biomass (including reserves). Dots, vertical crosses (+) and diagonal crosses (x) are US-Ne1, US-Ne2 and US-Ne3 observations respectively. Solid line shows the fit using parameters from Osborne et al. (2015) ($\lambda = 0.4$, $\kappa' = 3.06$) and dashed line shows a tuned fit ($\lambda = 0.38$, $\kappa' = 3.43$). Only points up until the maximum stem biomass for that site in that year are plotted.

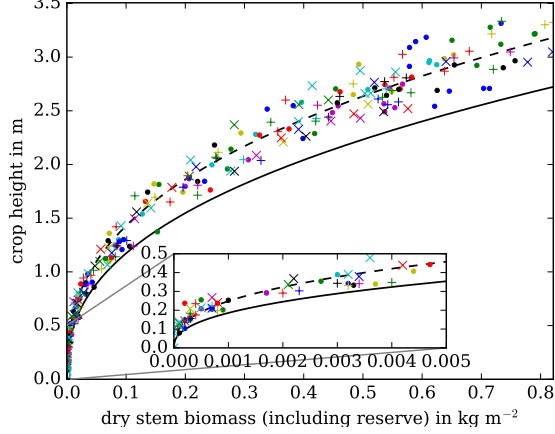





**Figure 6.** Specific leaf area against DVI. Dots, vertical crosses (+) and diagonal crosses (x) are US-Ne1, US-Ne2 and US-Ne3 observations respectively. Solid line shows the fit using parameters from Osborne et al. (2015) ($\gamma = 22.5$, $\delta = -0.2587$) and dashed line shows a tuned fit ($\gamma = 17.6$, $\delta = -0.33$).

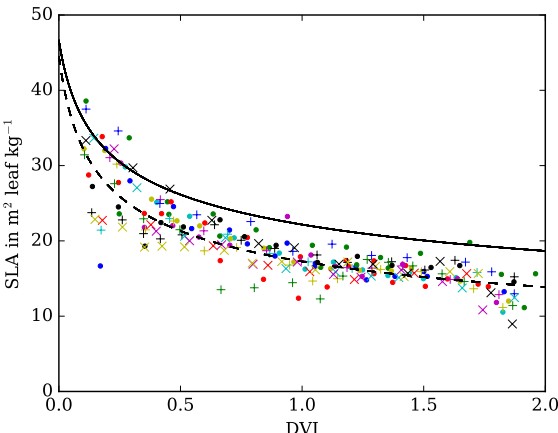

**Figure 7.** Carbon to biomass ratio in leaves against day after sowing. Dots, vertical crosses (+) and diagonal crosses (x) are US-Ne1, US-Ne2 and US-Ne3 observations respectively. The years 2001-4 are magenta, blue, cyan and yellow respectively. Solid black line shows the value used in Osborne et al. (2015), dashed black line shows the value used in this analysis.

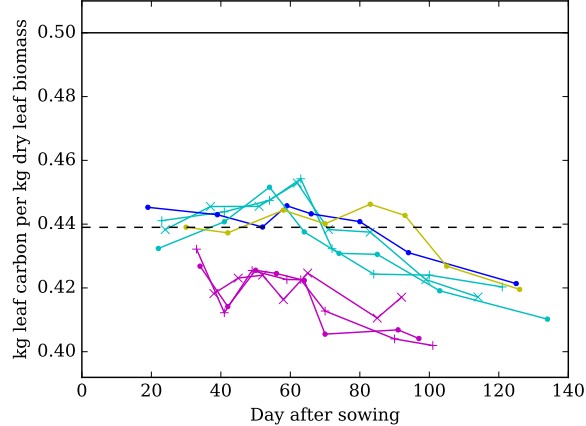





**Figure 8.** Above-ground biomass measurements against DVI. Dots, vertical crosses (+) are US-Ne1 and US-Ne2 observations respectively. Points from US-Ne3 are not shown. Intersection of the solid black line shows the initialisation used in Osborne et al. (2015), intersection of dashed black line shows the initialisation used in this study.

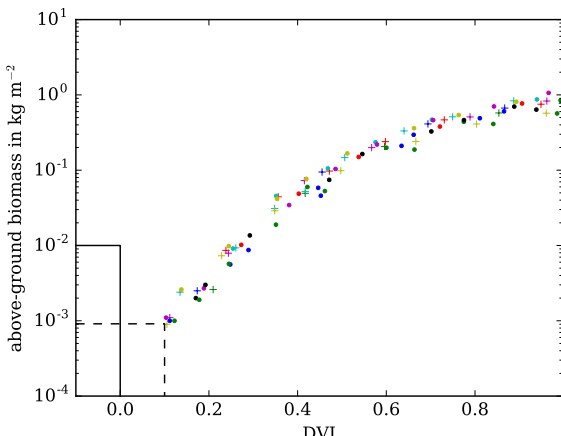

**Figure 9.** Yield fraction against the sum of the biomass in the reproductive parts of the maize crop (kernel, cob, husk, ear shank and silk) and the yellow leaf biomass, using the last measurement of the season. Dots, vertical crosses (+) and diagonal crosses (x) are US-Ne1, US-Ne2 and US-Ne3 observations respectively. Solid black line shows the value used implicitly in Osborne et al. (2015) and dashed black line shows the new, tuned value.

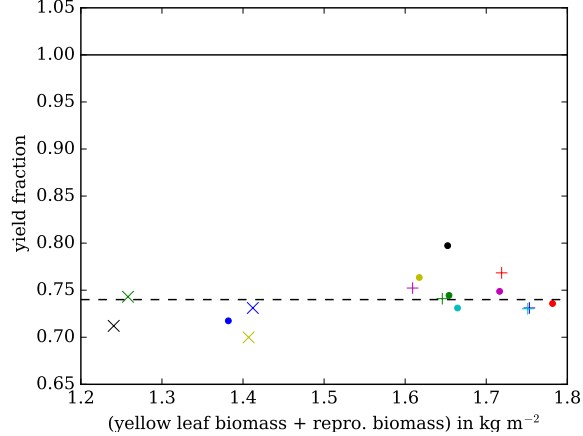





**Figure 10.** Stem biomass against the product of height and LAI (left) and stem biomass against the LAI (right). Dots, vertical crosses (+) and diagonal crosses (x) are US-Ne1, US-Ne2 and US-Ne3 observations respectively. Solid line shows the fit using the natural PFT parameters from Osborne et al. (2015) and dashed line shows a tuned fit using the relations for natural vegetation described in Section 2.2. Only points up until the maximum LAI measurement for that site in that year are shown.

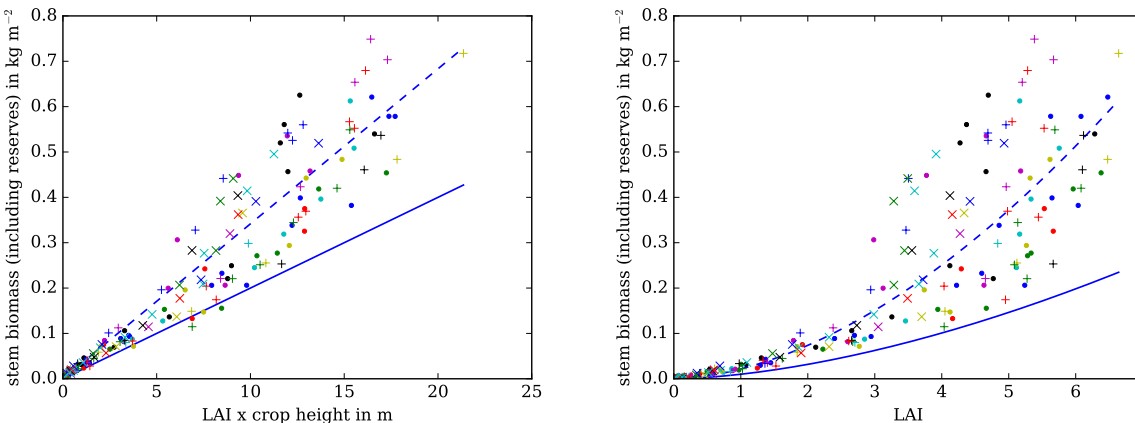

**Figure 11.** Height against stem biomass. Dots, vertical crosses (+) and diagonal crosses (x) are US-Ne1, US-Ne2 and US-Ne3 observations respectively. Solid line shows the fit using natural PFT parameters from Osborne et al. (2015) and dashed line shows a tuned fit using the relations for natural vegetation described in Section 2.2. Only points up until the maximum stem biomass for that site in that year are plotted.

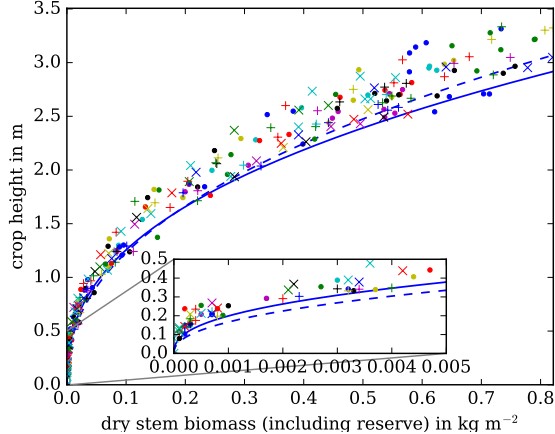





**Figure 12.** Balanced LAI (calculated from canopy height) against LAI. Dots, vertical crosses (+) and diagonal crosses (x) are US-Ne1, US-Ne2 and US-Ne3 observations respectively. Red: uses natural PFT parameters from Osborne et al. (2015), blue: uses new, tuned parameters. Dotted line shows $x = y$. Only points up until the maximum LAI measurement for that site in that year are shown.

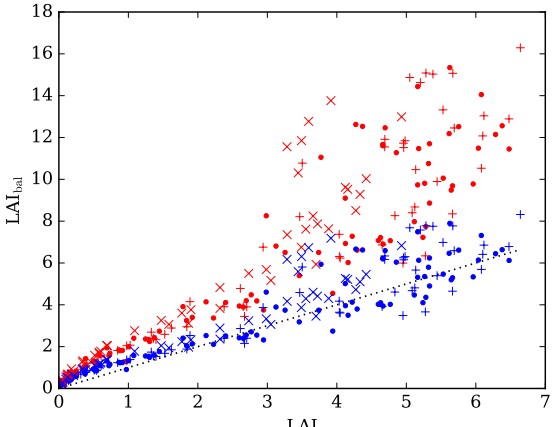





**Figure 13.** FAPAR against interpolated LAI observations. Top row uses FAPAR observations, while middle row and bottom rows use model FAPAR with $a = 1$ and $a = 0.65$ respectively, using observed LAI and diffuse radiation fractions. Dots, vertical crosses (+) and diagonal crosses (x) show US-Ne1, US-Ne2 and US-Ne3 respectively and all data are between emergence (DVI=0) and flowering (DVI=1). Colours show the cosine of the zenith angle.





**Figure 14.** Derived value of the clumping factor $a$ against LAI for each combination of FAPAR and observed diffuse radiation fraction. Dots, vertical crosses (+) and diagonal crosses (x) use US-Ne1, US-Ne2 and US-Ne3 LAI observations respectively. Colours show the cosine for the zenith angle (for legend, see Figure 13). Solid black line indicates $a = 1$ and dashed black line indicates $a = 0.65$.

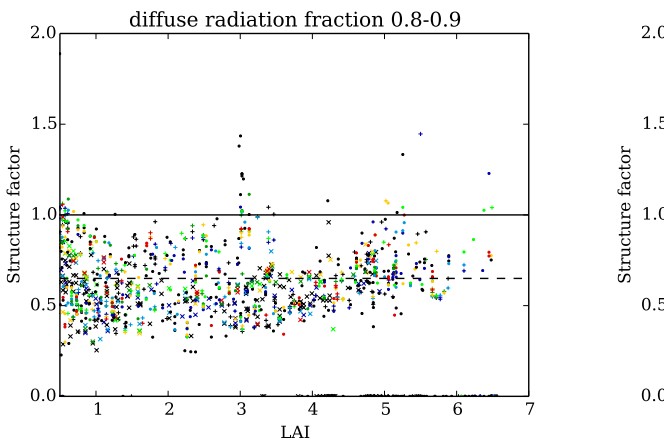

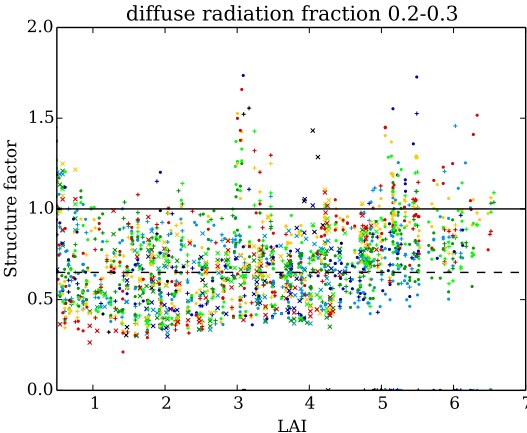

**Figure 15.** Observed ratio of nitrogen mass to carbon mass in leaves (left) and leaf nitrogen per leaf area (right) against day after sowing. Dots, vertical crosses (+) and diagonal crosses (x) are US-Ne1, US-Ne2 and US-Ne3 respectively. The years 2001-4 are magenta, blue, cyan and yellow respectively. $k_{nl} = 0$ i.e. leaf properties are assumed constant through the canopy.

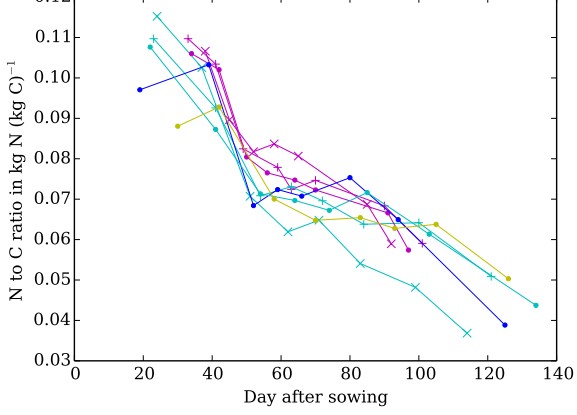

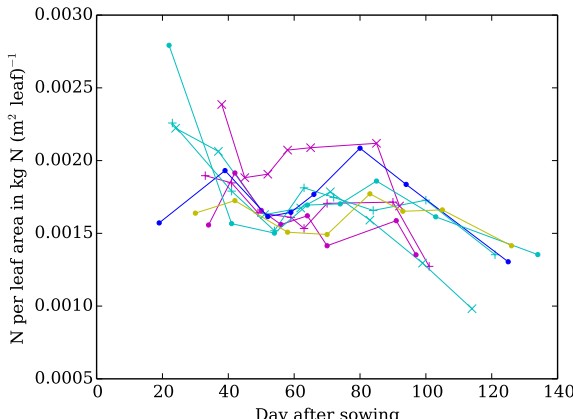





**Figure 16.** Observed leaf nitrogen per leaf area at top of canopy against day after sowing assuming a decay through the canopy with decay constant $k_{nl} = 0.2$. Dots, vertical crosses (+) and diagonal crosses (x) are US-Ne1, US-Ne2 and US-Ne3 respectively. The years 2001-4 are magenta, blue, cyan and yellow respectively.

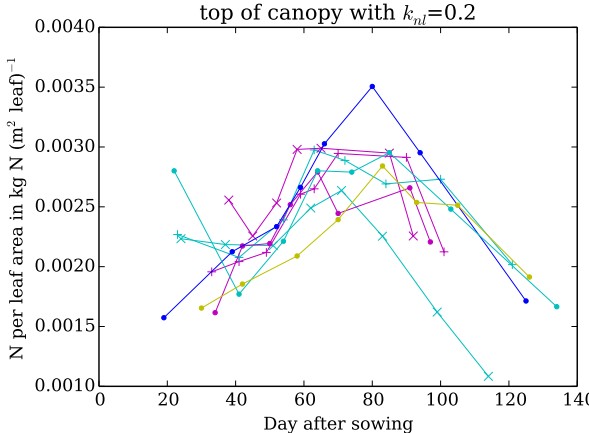

**Figure 17.** Parametrisations of $V_{\mathrm{cmax}}$ against leaf temperature. Solid black line shows default C4 grass in JULES. Dotted line shows the parameterisation for maize given in de Vries et al. (1989), black dashed line shows a fit to this using the JULES parameterisation. Blue dot-dashed shows the parameterisation for maize in Massad et al. (2007).

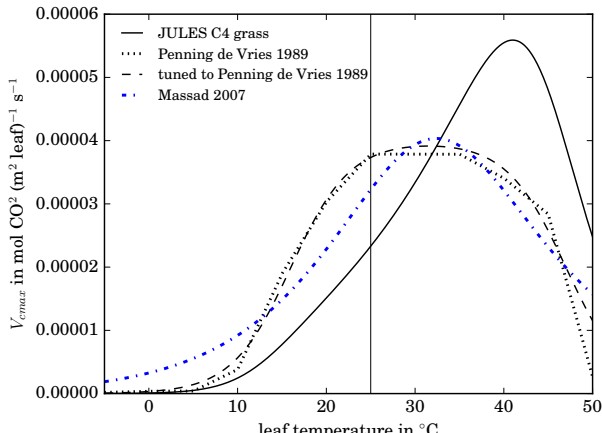





**Figure 18.** Time series of GPP for irrigated maize at the Mead FLUXNET sites US-Ne1 and US-Ne2. Blue: model, green: observations. JULES runs have the crop model switched off, LAI and canopy height prescribed and the input parameters in Table 1, Table 2, Table 3 and Table 4.





**Figure 19.** GPP (in $\mu$mol CO$_2$(m$^2$ground)$^{-1}$s$^{-1}$) against absorbed PAR (in $\mu$mol photons (m$^2$ground)$^{-1}$s$^{-1}$) for the hourly FLUXNET data (left) and hourly output from the model runs (right). LAI is between 3.5 and 4.5 and all points have DVI less than 1. Dots, vertical crosses (+) and diagonal crosses (x) indicate US-Ne1, US-Ne2 and US-Ne3 respectively. Colour: diffuse radiation fraction. JULES runs have the crop model switched off, LAI and canopy height prescribed and the input parameters in Table 1, Table 2, Table 3 and Table 4.

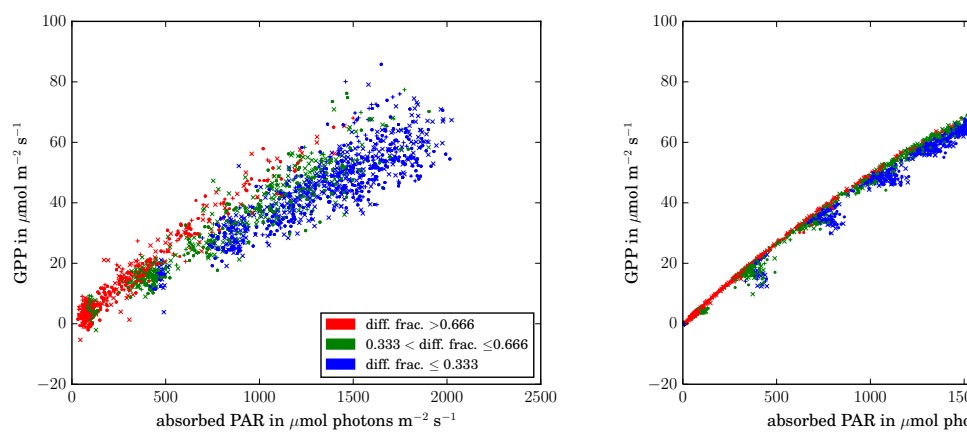

Plots of modelled GPP (blue) against observed GPP (green) are shown in Figure 18 for years in which irrigated maize was grown at the Mead FLUXNET sites US-Ne1 and US-Ne2. While the overall shape of the plots is good, it is clear that GPP in the model is significantly overestimated after the mid-season peak in observed GPP (corresponding to where LAI declines as the crop leaves senesce). As discussed earlier, the model $V_{\mathrm{cmax}}$ at a certain temperature stays constant whereas in reality it

would decline over the crop season. Implementing this decline into JULES would result in a much closer fit between the model GPP and observed GPP.

To a lesser extent, there also appears to be an overestimation of GPP in the model before senescence. This was investigated in more detail by comparing plots of FLUXNET GPP against observed APAR with plots of model GPP against APAR, for hourly measurements before the crop reaches DVI=1, for LAI bins of size 1 for all sites. Figure 19 shows the LAI bin 3.5 to 4.5.

There is a clustering of points due to the hourly resolution of the data, which is most clearly seen in the model output. Hours with high diffuse radiation fractions (red) are similar in both the FLUXNET data and the model output, although the scatter in the FLUXNET data is higher, as expected from the plots of observed FAPAR (Figure 13). For lower diffuse radiation fractions in the model, GPP decreases due to a combination of the effect of sunflecks and an increase in the effective decay constant of absorbed PAR through the canopy at the beginning and end of the day. Even when the scatter in the FAPAR observations is

taken into account, the decrease in GPP for lower diffuse radiation fractions does not appear to be as large in the model as in the GPP observations, and this is the source of the overestimation of GPP we saw in the model output in Figure 18 before the onset of senescence.

This effect was investigated further by considering the dependence on air temperature and vapour pressure deficit in the FLUXNET GPP data. As expected, the lower temperature points (Figure 20, top left) and lower VPD points (Figure 20, top





right) are clustered at low values of APAR. However, there does not seem to be a dependence on temperature or VPD at a constant APAR across the range of GPP observations.

Soil moisture stress is a factor that we have neglected in our runs, which could, if implemented, reduce GPP when the soil moisture is low. However, as Figure 20 shows for soil moisture content at a depth of 10cm (bottom left) and 25cm (bottom
right), at higher APAR values, points below a threshold of 30% appear to be distributed evenly across the range of GPP observations for a constant APAR.

Including a decrease in leaf nitrogen concentration through the canopy would have the effect of making the light use of the plant more efficient, which would increase model GPP still further. Increasing $V_{\mathrm{cmax,norm}}$ would have the effect of reducing model GPP at higher APAR values, but this would not solve the issue at mid-range APAR points $\sim 800$ $\mu$mol photons
$(\mathrm{m^2ground})^{-1}\mathrm{s}^{-1}$ and would also worsen the fit of the points with high diffuse radiation fractions.

It is therefore difficult to see a clear way in which the model parameter settings or processes should be improved. It would be possible to improve the validation against observations by decreasing $\alpha$ or changing the curvature parameter in the non-rectangular hyperbola implemented for light response within JULES (currently hard-wired) but it is difficult to justify this theoretically.

### 4.1.2 Respiration

The results from the model runs without the crop model can also be used to test the parameterisation of respiration.

Using a number of assumptions, the measurements from Mead can be used to get an approximate value for leaf maintenance respiration. First, approximate values for NPP were obtained by linearly interpolating the Mead above-ground biomass measurements to get a daily time series, and then differentiating. The fraction of NPP directed to the roots at each DVI was
calculated from the expression for maize in de Vries et al. (1989) (plotted in Figure 2) and then used to obtain the total NPP. Combining these NPP values with the GPP observations and assuming a value for the growth respiration coefficient of $r_g = 0.25$ and summing over the crop season leads to an estimation of the plant maintenance respiration $R_{pm}$. It is necessary to sum over the whole season, since the NPP and GPP calculated in this way appear to be slightly out of step with each other, and this effect dominates the daily time series of derived maintenance respiration.

The interpolated carbon pool observations were used to calculate the factor $\left(1 + \mu_{rl}\frac{C_{\mathrm{root}}}{C_{\mathrm{leaf}}} + \mu_{sl}\frac{C_{\mathrm{stem}}}{C_{\mathrm{leaf}}}\right)$ that converts between the leaf maintainance respiration and the total plant maintenance respiration. Note that the stem carbon observations had to be corrected using $\tau$ to get $C_{\mathrm{stem}}$. This factor was used to convert the leaf maintenance respiration outputted by the model to the total plant maintenance respiration.

Figure 21 shows the $R_{pm}$ derived from observed GPP against $R_{pm}/f_{dr}$ derived from the outputted model leaf maintenance
respiration. The x-axis therefore is independent of $f_{dr}$, which can be obtained from the gradient. Data from 2010 is not included (since the crop was damaged by hail). Both the default JULES C4 grass $f_{dr}$ (solid line) and the $f_{dr}$ used in our maize configuration (dashed line) are shown. It can clearly be seen that the new maize $f_{dr}$ is a better fit than the default C4 grass value. While there are many model and parameter assumptions ($r_g$, $\mu_{rl}$, $\mu_{sl}$, $\beta = 1$, $C_{\mathrm{root}}$, $\tau$) that have gone into this plot, this is still an important consistency check of our parameters.



**Figure 20.** Hourly FLUXNET GPP data (in $\mu\mathrm{mol}\,CO_2(m^2\,\mathrm{ground})^{-1}s^{-1}$) against observed APAR (in $\mu\mathrm{mol}\,\mathrm{photons}(m^2\,\mathrm{ground})^{-1}s^{-1}$). LAI is between 3.5 and 4.5 and all points have DVI less than 1. Dots, vertical crosses (+) and diagonal crosses (x) are US-Ne1, US-Ne2 and US-Ne3 observations respectively. Colour indicates air temperature (top left), vapour pressure deficit (top right), soil water content at 10cm (lower left) and soil moisture content at 25cm (lower right).

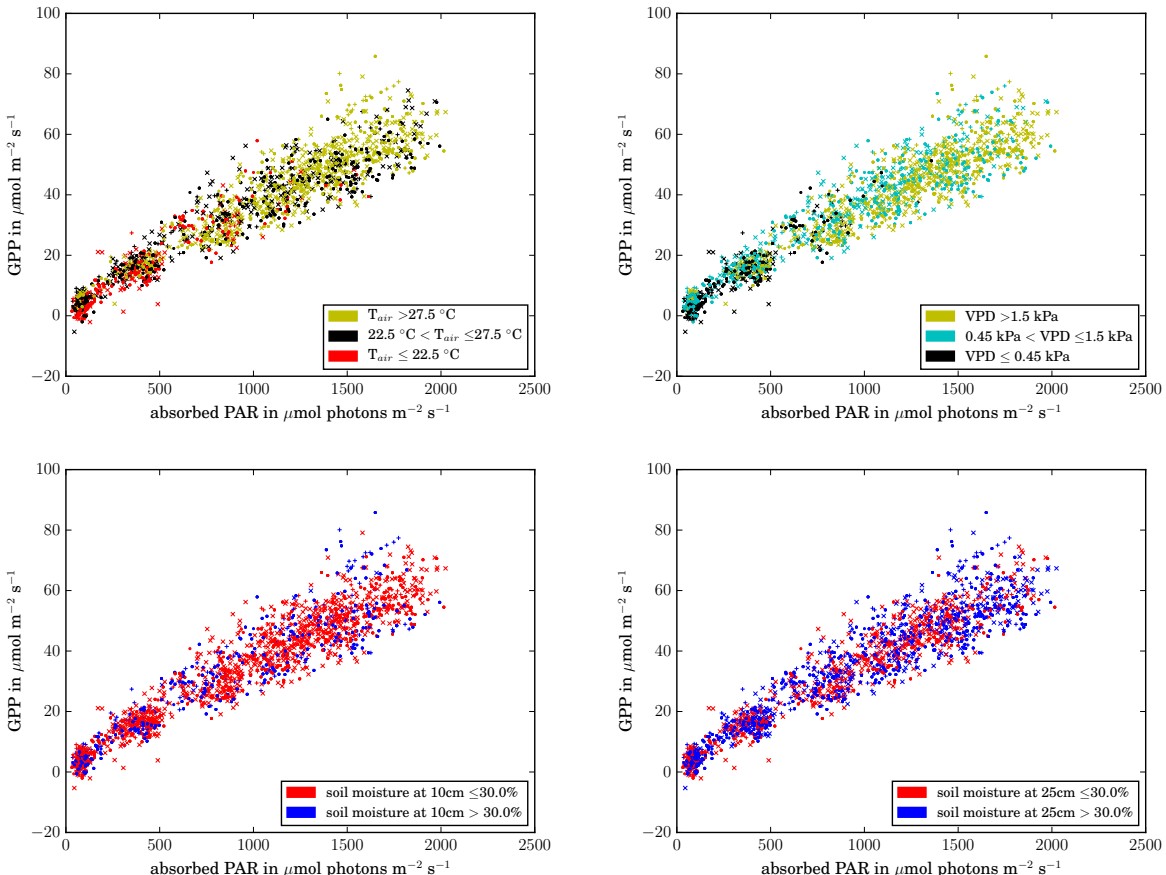

## 4.2 Results from JULES runs with the crop model

This section describes the results from the runs for the irrigated maize seasons from the Mead sites, with the crop model switched on and the parameter settings summarised in Table 1, Table 2, Table 3 and Table 4





**Figure 21.** $R_{pm}$ derived from observed GPP against $R_{pm}/f_{dr}$ derived from the outputted model leaf maintenance respiration. Dots and vertical crosses (+) are US-Ne1 and US-Ne2 respectively. JULES runs have the crop model switched off, LAI and canopy height prescribed and the input parameters in Table 1, Table 2, Table 3 and Table 4. Black lines pass through the origin and have gradient 0.025 (solid line) and 0.0096 (dashed line), corresponding to the value of $f_{dr}$ used in Osborne et al. (2015) and this study respectively.

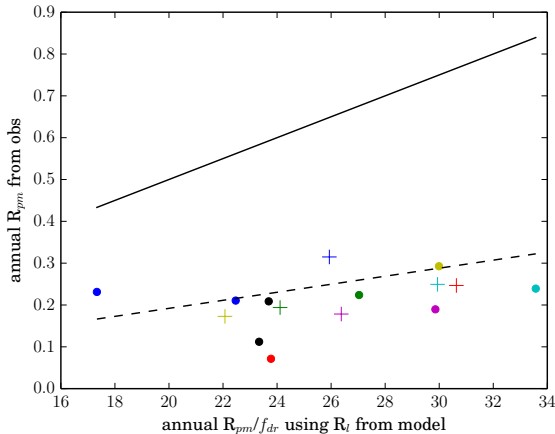





**Figure 22.** Time series of GPP for irrigated maize at the Mead FLUXNET sites US-Ne1 and US-Ne2. Blue: model, green: observations. JULES runs have the crop model switched on and the input parameters in Table 1, Table 2, Table 3 and Table 4.





**Figure 23.** Time series of LAI for irrigated maize at the Mead FLUXNET sites US-Ne1 and US-Ne2. Blue: model, red: observations. JULES runs have the crop model switched on and the input parameters in Table 1, Table 2, Table 3 and Table 4.





**Figure 24.** Time series of LAI for irrigated maize at the Mead FLUXNET sites US-Ne1 and US-Ne2. Blue: model, red: observations. JULES runs have the crop model switched on and $\gamma=18.0$ and $\delta=-0.45$. All other input parameters are as described in Table 1, Table 2, Table 3 and Table 4.

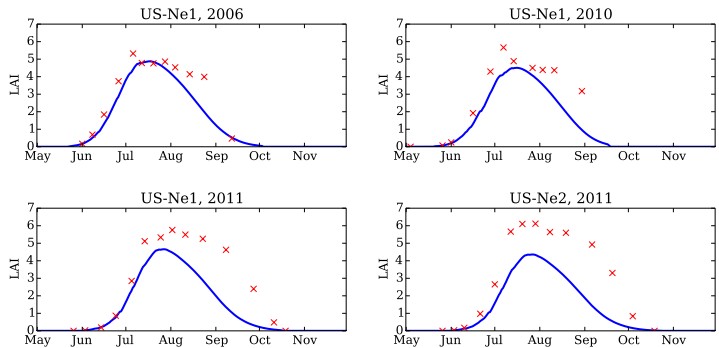

Figure 22 compares the model GPP and the observations, and shows very close agreement. This is influenced by a cancellation of two effects: as identified in the previous section, the GPP per APAR in the model is biased high, whereas the outputted LAI is biased low, shown in Figure 23. In part the reduction in modelled LAI compared to observations was deliberately introduced when tuning the senescence parameters, so that a quicker decrease in LAI compensates partially for the model not including a decrease in leaf photosynthetic capacity. However, in some years (2006, 2010, 2011 for US-Ne1 and 2011 for US-Ne2), the LAI is too small in the crop season up to anthesis. This is due to the high sensitivity of the plant in its early life to parameter settings, due to the feedback between NPP and LAI. In these site and year combinations (2006, 2010, 2011 for US-Ne1 and 2011 for US-Ne2), temperatures between DVI 0.1 and DVI 0.2 are higher on average, and so DVI is increasing more rapidly, which gives the plant less time to accumulate NPP, leading to a reduced rate of increase of LAI with respect to DVI in the model runs at this growth stage. On the other hand, the SLA observations for these years in the early crop season are particularly high compared to the rest of the distribution, which means that the observations do not show this reduced rate of increase of LAI at this growth stage. Fitting $\gamma$ and $\delta$ to the SLA observations in just these site and year combinations (2006, 2010, 2011 for US-Ne1 and 2011 for US-Ne2) gives 18.0 and -0.45 respectively. Using these parameters in JULES runs with the crop model gives much better agreement with LAI observations (Figure 24). This is also consistent with the result from US-Ne2 in 2010: since the crop emerges 9 days after the crop in US-Ne1, the period of relatively high temperatures mostly falls before the crop is initialised. It is possible that parameterising SLA with day after emergence rather than with DVI might improve the fit between model and observed LAI by reducing the sensitivity of the SLA parameterisation to temperature.

The canopy height is well represented in the runs (Figure 25). The above-ground carbon in the model also fits the observations well (Figure 26). The harvest carbon pool (which includes the reproductive parts of the plant and the yellow leaves) is overestimated in the model, which is consistent with the overestimation of GPP during the senescence period.





**Figure 25.** Time series of canopy height for irrigated maize at the Mead FLUXNET sites US-Ne1 and US-Ne2. Blue: model, red: observations. JULES runs have the crop model switched on and the input parameters in Table 1, Table 2, Table 3 and Table 4.





**Figure 26.** Time series for above-ground carbon for irrigated maize at the Mead FLUXNET sites US-Ne1 and US-Ne2. Blue: model, red: observations. JULES runs have the crop model switched on and the input parameters in Table 1, Table 2, Table 3 and Table 4.





**Figure 27.** Time series of the carbon in the harvest pool (reproductive parts of the crop and yellow leaves). Irrigated maize at the Mead FLUXNET sites US-Ne1 and US-Ne2. Blue: model, red: observations. JULES runs have the crop model switched on and the input parameters in Table 1, Table 2, Table 3 and Table 4.



## 5 Conclusions

The JULES-crop parametrisation of crops within JULES was introduced to improve the carbon and energy fluxes in the model over croplands and to investigate the effect of weather and climate on food and water resources, at global, regional and local scales. In this evaluation paper, we have looked in detail at how the input parameters can be tuned for one crop - maize - at one location - Mead, US - where there are a wide variety of observations to probe how the model components perform, both separately and in combination.

In previous analyses with JULES-crop, it has been assumed that model photosynthesis and respiration parameters can be set to the default C3 grass values for C3 crops and the default C4 grass values for C4 crops. We have shown that a significant improvement can be made when modelling irrigated maize if these parameters are tuned to results from the literature for maize. We have also improved the parameters required in the crop-model part of JULES (such as partition fractions and allometric constants) by tuning directly to observations.

With the new parameters, there is good agreement between modelled GPP and observed GPP up until anthesis if the feedback between NPP and LAI is removed by switching the crop model off and prescribing LAI (and canopy height) when the skies are mostly overcast. The model tends to overestimate GPP for clearer skies. After anthesis, there is a much greater overestimation of GPP, due to the model being unable to capture the decrease in photosynthetic capability at the leaf level over time in the crop. The respiration parameters were more difficult to test in isolation, but integrating model respiration over the entire crop season produced results that were consistent with the GPP and carbon pool observations.

Running the full crop model, including all the new parameters, produced GPP time series that were very close to the observations. This was helped partially by a cancellation of two biases - the model GPP for a certain LAI was biased high, as we have just discussed, and the LAI in the model was biased low compared to the observations. There were a few anomalous years in which the peak LAI in the model was approximately two thirds of the peak LAI in the observations, which may imply oversensitivity to initial conditions. The amount of above-ground carbon was reproduced well, although the amount of carbon in the harvest pool was overestimated in most cases.

There should be three main priorities for extending this work to improve the representation of maize at these sites. Firstly, work should be done to tune the parametrisation of soil moisture stress of maize, so that the water balance of the irrigated sites could be accurately modelled and runs for the non-irrigated site could also be included. This configuration would be applicable for a wider set of situations, for example, the ability to model the yield of non-irrigated crops would be a necessary requirement for addressing food security questions. Secondly, a parametrisation of the maximum rate of carboxylation of Rubisco $V_{\mathrm{cmax}}$ should be added that allows it to vary over the course of the crop season. Thirdly, these runs have been tightly constrained by observed sowing, emergence, flowering and harvest dates. For most regions, and for any climate projections, this sort of data will not be available. Therefore, it would be necessary to investigate the performance of the model in individual years at these sites when given generic values for the thermal time parameters.

While this study has been carried out at one particular location, the JULES input parameters for individual crop tiles are designed so that they fully characterise a particular crop variety. Therefore, the logical next step, after addressing the issues





described above, would be to test this model feature by investigating how well these new parameters perform at other sites which grow the same variety. To scale this up from the site level to global applications, it is important to consider that there is a limit to the number of crop tiles that can be simulated in one run, due to data and computational resource constraints. Therefore each variety within a crop type cannot be simulated as a separate tile, with its own individual set of parameters.

Previous JULES-crop global analyses, such as in Osborne et al. (2015), have therefore aimed at a compromise: four main crop types were considered (maize, soybean, wheat and rice), and each of these crop types were given spatially-varying thermal times between emergence and flowering and between flowering and harvest, which were tuned to observed growing season lengths. The other parameters were set to 'generic' parameter values for that crop type. In this way, some of the variation between varieties within a crop type was captured. The parameters given in Table 2 and Table 3 are a good starting point for a

generic 'maize' tile, but would need to be tested at locations with different varieties, chosen for different climatic conditions, to determine whether they are able to sufficiently capture the characteristics of maize on a global scale to be able to address useful scientific questions. If not, the use of more than one 'maize' tile should be considered, or key parameters should be identified which could be allowed to vary spatially.

## 6 Code availability

This study uses JULES revision 5061, which is between the 4.6 and 4.7 releases. The code can be downloaded from the JULES FCM repository at

`https://code.metoffice.gov.uk/trac/jules/`

(registration required).

The pySellersTwoStream package is available at

20 `https://github.com/tquaife/pySellersTwoStream`.

## 7 Data availability

Unless otherwise noted, all site observations discussed in this paper was obtained from the Site Information pages of the AmeriFlux website (`http://public.ornl.gov/ameriflux/` or personal communication with the Mead sites Research Technologist.

## 8 Copyright



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

*Acknowledgements.* KW gratefully acknowledges financial support from the European Commission's 7th Framework Programme for Research (EU/FP7) under Grant Agreement 308291 (EUPORIAS) and 603864 (HELIX). AW was supported by EU/FP7 under Grant Agreement 603542 (LUC4C) and DH was supported by the Joint UK BEIS/Defra Met Office Hadley Centre Climate Programme (GA01101). TQ was funded by the NERC National Centre for Earth Observation, UK. AH was supported by an EPSRC Living With Environmental Change fellowship No EP/N030141/1. We acknowledge the following AmeriFlux sites for their data records: US-Ne1, US-Ne1, US-Ne3. In addition, funding for AmeriFlux data resources and core site data was provided by the U.S. Department of Energy's Office of Science. The authors would like to thank Camilla Mathison and Alberto Martinez de la Torre for useful discussions.



**Table 1.** JULES flags.

|  | Osborne et al. (2015) | This study | Discussion |
|---|---|---|---|
| `can_rad_mod` | 5 (6 was not available) | 6 | Recommended option for layered canopy in version 4.6 |
| `l_irrid_dmd` | F∗ | T | Section 2.6 |
| `irr_crop` | - | 0 | Section 2.6 |
| `l_trait_phys` | F∗ | F | Section 3.5.2 |
| `l_scale_resp_pm` | F∗ | T | Section 2.5 |
| `l_leaf_n_resp_fix` | F∗ | - | Bug fix, affects `can_rad_mod`=5 but not `can_rad_mod`=6 |
| `l_prescsow` | T | T | Section 3.3.1 |
| `l_phenol` | T,F | F | Only relevant in non-crop runs |

∗ parameter was hard-wired

**Table 2.** JULES plant functional type parameters extended to represent maize.

| | | Osborne et al. (2015) | This study | Discussion |
|---|---|---|---|---|
| $c3$ | c3_io | 0 | 0 | Maize is a C4 plant. |
| $d_r$ | rootd_ft_io | 0.5 | 1.7 | Not important in irrigated runs. |
| $dq_{\mathrm{crit}}$ | dq_crit_io | 0.075 | 0.075 | Section 3.5.2 |
| $f_d$ | fd_io | 0.025 | 0.0096 | Section 3.5.3 |
| $f_0$ | f0_io | 0.8 | 0.4 | Section 3.5.2 |
| $n_{\mathrm{eff}}$ | neff_io | $4.00 \times 10^{-4}$ | $5.7 \times 10^{-4}$ | Section 3.5.2 |
| $n_l(0)$ | nl0_io | 0.06 | 0.07 | Section 3.5.2 |
| $T_{\mathrm{low}}$ | tlow_io | 13.0 | 16.0 | Section 3.5.2 |
| $T_{\mathrm{upp}}$ | tupp_io | 45.0 | 47.0 | Section 3.5.2 |
| $k_n$ | kn_io | 0.78 | - | - |
| $k_{nl}$ | knl_io | - | 0.0 | Section 3.5.2 |
| $Q_{10,\mathrm{leaf}}$ | q10_leaf_io | 2.0 | 1.0 | Section 3.5.2 |
| $\mu_{rl}$ | nr_nl_io | 1.0 | 0.39 | Section 3.5.3 |
| $\mu_{sl}$ | ns_nl_io | 1.0 | 0.43 | Section 3.5.3 |
| $r_g$ | r_grow_io | 0.25 | 0.25 | Section 3.5.3 |
| | orient_io | 0 | 0 | Section 3.5.1 |
| $\alpha$ | alpha_io | 0.06 | 0.055 | Section 3.5.2 |
| $\omega_{\mathrm{PAR}}$ | omega_io | 0.17 | 0.17 | Section 3.5.1 |
| $\alpha_{\mathrm{PAR}}$ | alpar_io | 0.1 | 0.1 | Section 3.5.1 |
| | fsmc_mod_io | 0* | 1 | Not important in irrigated runs. |
| | fsmc_p0_io | 0.0* | 0.65 | Not important in irrigated runs. |
| | | | | This value is consistent with e.g. |
| | | | | Ray et al. (2002), Bänziger et al. (2000). |
| $a$ | can_struct_a_io | 1.0* | 0.65 | Section 3.5.1 |
| $a_{ws}$ | a_ws_io | 1.0 | 0.88 | Section 3.4 |
| $\eta_{sl}$ | eta_sl_io | 0.01 | 0.0170 | Section 3.4 |
| $a_{wl}$ | a_wl_io | 0.005 | $9.5 \times 10^{-3}$ | Section 3.4 |
| $b_{wl}$ | b_wl_io | 1.667 | 1.767 | Section 3.4 |
| $\sigma_l$ | sigl_io | 0.05 | 0.0244 | Section 3.4 |

∗ parameter was hard-wired





**Table 3.** Values of the crop-specific JULES parameters used to represent maize.

|  |  | Osborne et al. (2015) | This study | Discussion |
|---|---|---|---|---|
| $T_{\mathrm{b}}$ | t_bse_io | 281.15 | 281.15 | Section 3.3.1 |
| $T_{\mathrm{o}}$ | t_opt_io | 303.15 | 303.15 | Section 3.3.1 |
| $T_{\mathrm{m}}$ | t_max_io | 315.15 | 315.15 | Section 3.3.1 |
| $\mathrm{TT_{emr}}$ | tt_emr_io | 80 | Table 5 | Section 3.3.1 |
| $\mathrm{TT_{veg}}$ | tt_veg | Osborne et al. (2015) fig.3 | Table 5 | Section 3.3.1 |
| $\mathrm{TT_{rep}}$ | tt_veg | Osborne et al. (2015) fig.3 | Table 5 | Section 3.3.1 |
| $P_{\mathrm{sen}}$ | pp_sens_io | 0.0 | 0.0 | DVI assumed to have no photoperiod dependence |
| $P_{\mathrm{crit}}$ | crit_pp_io | 24 | - | Not used when pp_sens_io=0.0 |
| $r_{\mathrm{dir}}$ | rt_dir_io | 0.0 | 0.0 | Not important in irrigated runs. |
| $\alpha_{\mathrm{root}}$ | alpha1_io | 13.5 | 13.5 | Section 3.3.2 |
| $\alpha_{\mathrm{stem}}$ | alpha2_io | 12.5 | 12.1 | Section 3.3.2 |
| $\alpha_{\mathrm{leaf}}$ | alpha3_io | 13.0 | 13.1 | Section 3.3.2 |
| $\beta_{\mathrm{root}}$ | beta1_io | $-15.5$ | -15.0 | Section 3.3.2 |
| $\beta_{\mathrm{stem}}$ | beta2_io | $-12.5$ | -12.1 | Section 3.3.2 |
| $\beta_{\mathrm{leaf}}$ | beta3_io | $-14.0$ | -14.1 | Section 3.3.2 |
| $\gamma$ | gamma_io | 22.5 | 17.6 | Section 3.3.6 |
| $\delta$ | delta_io | $-0.2587$ | -0.33 | Section 3.3.6 |
| $\tau$ | remob_io | 0.35 | 0.12 | Section 3.3.3 |
| $f_{C,\mathrm{root}}$ | cfrac_r_io | 0.5 | 0.439 | Not important in irrigated runs. |
| $f_{C,\mathrm{stem}}$ | cfrac_s_io | 0.5 | 0.439 | Section 3.3.7 |
| $f_{C,\mathrm{leaf}}$ | cfrac_l_io | 0.5 | 0.439 | Section 3.3.7 |
| $\kappa$ | allo1_io | 3.5 | 3.6 | Section 3.3.5 |
| $\lambda$ | allo2_io | 0.4 | 0.38 | Section 3.3.5 |
| $\mu$ | mu_io | 0.05* | 0.02 | Section 3.3.4 |
| $\nu$ | nu_io | 0.0* | 4.0 | Section 3.3.4 |
| $\mathrm{DVI_{sen}}$ | sen_dvi_io | 1.5* | 0.4 | Section 3.3.4 |
| $C_{\mathrm{init}}$ | initial_carbon_io | 0.01* | $8.0 \times 10^{-4}$ | Section 3.3.8 |
| $\mathrm{DVI_{init}}$ | initial_c_dvi_io | 0.0* | 0.1 | Section 3.3.8 |
| $T_{\mathrm{mort}}$ | t_mort_io | t_bse_io * | 273.15 | Section 3.3.1 |
| $f_{\mathrm{yield}}$ | yield_frac_io | 1.0* | 0.74 | Section 3.3.9 |

∗ parameter was hard-wired





**Table 4.** Other relevant JULES parameter values.

|  | Osborne et al. (2015) | This study | Discussion |
|---|---|---|---|
| `diff_frac` | 0.0 | hourly observations | Section 3.2 |
| `co2mmr` | $5.241 \times 10^{-4}$ (JULES default) | annual observations from Dlugokencky and Tans (2016) | Section 3.2 |

**Table 5.** Thermal times in degree days based on crop dates recorded at the Mead FLUXNET sites, combined with hourly observed temperatures.

| year | sowing DOY | sowing-emergence | emergence-flowering | flowering-maturity | flowering-harvest | sowing-harvest |
|---|---|---|---|---|---|---|
| US-Ne1 | | | | | | |
| 2002 | 130 | 85.55 | - | - | - | 2011 |
| 2003 | 135 | 59.71 | 868.6 | - | 1001 | 1938 |
| 2004 | 125 | 113.0 | 844.1 | 784.7 | 977.0 | 1945 |
| 2005 | 124 | 107.3 | 923.2 | 869.4 | 1083 | 2129 |
| 2006 | 124 | 59.32 | 819.8 | 883.6 | 1086 | 1973 |
| 2007 | 121 | 84.84 | 865.7 | 932.6 | 1331 | 2281 |
| 2008 | 120 | 64.48 | 888.4 | 967.3 | 1138 | 2102 |
| 2009 | 110 | 89.44 | 903.6 | 836.2 | 959.3 | 1961 |
| 2010 | 109 | 84.62 | 808.3 | 935.5 | 1011 | 1917 |
| 2011 | 137 | 69.71 | 819.9 | 827.1 | 980.1 | 1885 |
| 2012 | 114 | 58.90 | 718.3 | 961.6 | 1275 | 2062 |
| US-Ne2 | | | | | | |
| 2003 | 134 | 53.41 | 830.0 | - | 1005 | 1887 |
| 2005 | 122 | 95.51 | 822.8 | 923.9 | 1218 | 2153 |
| 2007 | 121 | 96.33 | 849.6 | 932.3 | 1300 | 2254 |
| 2009 | 111 | 93.63 | 853.0 | 801.6 | 884.72 | 1837 |
| 2010 | 110 | 108.2 | 846.2 | - | 908.4 | 1874 |
| 2011 | 137 | 67.10 | 792.0 | 864.3 | 1039 | 1912 |
| 2012 | 115 | 55.92 | 694.1 | 993.0 | 1282 | 2042 |
| US-Ne3 | | | | | | |
| 2003 | 133 | 58.81 | 802.0 | - | 999.9 | 1870 |
| 2005 | 116 | 85.06 | 922.0 | 875.0 | 1163 | 2178 |
| 2007 | 122 | 117.66 | 806.0 | 919.5 | 1329 | 2263 |
| 2009 | 112 | 90.88 | 820.1 | 786.8 | 1032 | 1953 |
| 2011 | 122 | 112.7 | 802.5 | 923.6 | 1125 | 2051 |