# Peer review of "Evaluation of JULES-crop performance against site observations of irrigated maize from Mead, Nebraska"

_Geoscientific Model Development, 2016_

## Referee Comment (RC1) · Anonymous Referee #1 · 25 Oct 2016

Improving the representation over croplands is indeed an important direction to a better Earth system model. Using JULES-crop and data from two eddy flux sites, the authors of this manuscript adjusted parameters for maize and examined the performance of JULES-crop over the sites. The presentation of the manuscript is detailed. However, some revisions are still needed in order to reflect the state-of-the-art understanding on the pros and cons of the model and its implication for the modelling community.

Crop phenology simulated by JULES-crop is still purely temperature driven. Many crop models have evolved to include impacts from other factors, such as precipitation, nutrient and day length. The authors should recognize biases it may bring in simulating crop phenology.

[Figure]

Many of the results have not been well presented and discussed. For example, Figure 23 not only shows that the JULES-crop has low bias in simulating LAI, but also shows that the capability of the model to capture interannual variability of LAI is very limited. Why is that? The high bias of GPP and low bias of LAI is intriguing. This should be further explored and explained because it appears implying that models might get good results with wrong reasons.

The discussion of scaling up from sites to the globe is too superficial. At least, there are tests that the authors can do to facilitate this discussion. For example, the authors can compare the site simulations against global simulations (Osborne et al. 2015) or the site simulations with parameters from. Osborne et al., (2015). This will give us better impression on the uncertainties JULES-crop has now for global simulations. Otherwise, I did not see the reason why this is the conclusion of this study.

It is also important to compare the efforts in JULES-crop with other land surface crop model, such as CLM-crop and ORCHIDEE-crop.
* * *

---

## Referee Comment (RC2) · Anonymous Referee #2 · 7 Nov 2016

The manuscript by Williams et al. describes and evaluates the performance of the JULES model with a new set of parameterizations for maize at the irrigated Mead, NE AmeriFlux site. The paper includes a detailed description of how site observations were used to optimize variables in JULES-crop, an evaluation of the model without crops, and an evaluation of the model with the new maize parameters. However, the manuscript requires some major revisions before it is ready for publication.

The central component of the paper is well written with a thorough description of the model and the parameter calibration, but the remainder of the manuscript lacks strong detail. For example, the introduction doesn't include much of a motivation for the study. What are the goals of the JULES crop model – yield, carbon, productivity and why is

parameterizing for the Mead site a valuable exercise? The results and conclusion are also fairly brief.

The authors mention in the abstract and introduction that they used observations from three MEAD FLUXNET sites, but they only use irrigated sites for model comparison? Why didn't they include the rainfed site in the analysis – especially when most cropland relies on rain to meet water demands?

In Section 2 and 3, some attention to the equations to define the parameters is needed to understand how the model works without reading other papers. The tables do not provide the necessary information to a non-user of JULES. See Technical Comments.

In Section 3, I would like some additional discussion on how the authors chose the parameters calibrated in this study. Was a sensitivity experiment done that indicated these parameters were important or were the parameters chosen in Tables 1-4 because they were convenient given the available observational data? If there was not a sensitivity study, perhaps the authors could highlight which parameters showed the most importance for the model results.

The authors compared the default JULES model without crops with the revised JULES crop model with updated parameters. This seems strange, what is the purpose of calibrating the LAI and height of JULES (without crops) with the newly parameterized JULES crop when a JULES-crop model already exists. I think it might also be more useful to compare the default JULES crop model from Osborne et al. (2015) with the newly revised site-specific parameterization. I also think it might be useful to look at the model performance at other sites.

It would also be useful to perform an uncertainty analysis of the parameters. This would be a valuable not only for the current model analysis, but also for extrapolating to other sites or globally. I'm not asking the authors to do this for this publication, but parameter uncertainty should be included in the discussion.

[Figure]

Technical Comments:

1. Eq. 1 has several parameters for temperature that aren't clearly defined (Table 3?). Relatedly, the second column in Tables 1-4 is not particularly useful to the reader; perhaps more appropriate would be a description for the variable rather than the model assigned parameter name.

2. Eq. 3: What is the difference between j and i? Again, it's not clear to a non-user of the model what those parameters are since they are not defined in the text or in the table.

3. P 35, L 19-20: Figure 27 should be referenced here.

4. Figure 26 & Figure 27: Why does US-Ne2 2010 have no observations?

---

## Referee Comment (RC3) · Anonymous Referee #3 · 15 Nov 2016

General comments:

The study describes and evaluates the new parameterization and improved coding added to JULES-crop since the original paper by Osborne (2015); it specifically evaluates the new parameterization used for maize crop over irrigated sites in Mead, Nebraska. The paper deals with an important study; however, it has certain major flaws.

Although the Abstract indicates that observations for maize at all three sites of Mead, Nebraska including one rain fed site (i.e. US-NE3) has been considered in deriving model parameters, according to the Introduction and the other sections, only the model performance for irrigated maize has been evaluated; some results from the rain fed maize might enhance the quality and validity of this study.

The JULES-crop simulations were run on crop tiles. At the US-NE2 site, soybean crop is also present in crop rotation on the same crop tile during even-numbered years. It would have been interesting to see how the model simulates crop-rotation on the same crop tile/s, as the paper also mentions that the model has been differently parameterized for C3 and C4 crops. This is very important, especially as the authors have plans for coupled-runs in the future, where the model should be able to simulate the carbon fluxes over a continuous time series. At least some mention/description on the model performance with regard to crop rotation needs to be included.

The paper is a bit too long; especially the number of figures is too large. Please try to reduce the number of figures, leaving only those that are essential and directly related to the predicted fluxes.

Specific comments:

Although the text on line 13-14 on p 35 mentions that the use of certain parameters improves the prediction of LAI, the improvement in the magnitude can be seen only during certain years when we compare the Figures 23 and 24. It seems that the model still needs improvement with regard to LAI, as the seasonality is not properly captured by the modeled LAI compared to the observed LAI.

The reader hardly can get any information from Tables 1-4; an additional column which describes each parameter listed in columns 1 and 2 might be helpful (or the model terms in column 2 could be replaced with easily understandable descriptors of each parameter). The column heading of the last column may be changed to 'Remarks' (instead of 'Discussion').

The paper does not seem to be in its final form, as still there are some typos and other errors, some of which are described below. So a thorough check on those is also needed.

P3 L13 'resp' should be replaced with 'respectively'

[Figure]

P5 L20 has 'a number options'. Please correct it.

P5 L24 Zenith angle dependence (of what)?

P6 L17 Q10 should be replaced with Q10

P14 L11 Instead of 'downloaded 15.09.2016', please provide a proper reference/web source.

P15 L24 (and everywhere else) 'Parametrisation' needs to be replaced with 'parameterization'.

P 20 Figure 4 has several lines in each color. Unless the authors explain what those are, the figure does not have much meaning to it (e.g. what are those several lines in black color mean? Which site does each of those correspond to?).

P 47 Heading of Table 5 mentions 'thermal units in degree days', whereas 'degree days' does not appear anywhere else in the text (According to P3 L1 crop development status is parameterized by a crop development index (DVI) which is determined by specific thermal time parameters set by the user (P3 L6-7). Degree days in Table 5 need to be related to the above description on p3.

————————————————

---

## Author Comment (AC1) · 16 Dec 2016

Please see the supplement for the response to each reviewer and the proposed new version of the manuscript (with changes marked).

Please also note the supplement to this comment: http://www.geosci-model-dev-discuss.net/gmd-2016-252/gmd-2016-252-AC1-supplement.pdf

---

## Author Response (AR1)

**Response to reviewers: Evaluation of JULES-crop performance against site observations of irrigated maize from Mead, Nebraska**

Karina Williams[1], Jemma Gornall[1], Anna Harper[2], Andy Wiltshire[1], Debbie Hemming[1], Tristan Quaife[3], Tim Arkebauer[4], and David Scoby[4]

[1]Met Office Hadley Centre, Exeter, UK
[2]College of Engineering, Mathematics, and Physical Sciences, University of Exeter, Exeter, UK
[3]National Centre for Earth Observation, Department of Meteorology, University of Reading, UK.
[4]Department of Agronomy and Horticulture, University of Nebraska-Lincoln, Lincoln, USA

*Correspondence to:* Karina Williams (karina.williams@metoffice.gov.uk)

**1   General notes to all reviewers**

We thank all the reviewers for their thorough reading of the manuscript and insightful and helpful comments. One thing that has become clear to us after reading the response from all three reviewers is that we have not described the aim of the paper with sufficient clarity. We will address this deficiency in the manuscript and thank the reviewers for drawing our attention to it.

5    Our purpose in this study is not to do a comprehensive evaluation of JULES-crop. JULES-crop is an existing crop parametri-sation available in the community land surface model JULES since June 2014, which has been in use across the JULES community in studies with a variety of different aims, spatial scales, geographical locations and crops. The original GMD discussion paper that presented this model, Osborne et al 2015, included a range of crop varieties and the performance at global and site scale.

10    Instead, this manuscript attempts to address a very particular question: how can one particularly rich dataset (irrigated maize grown at the research station in Mead) be used to probe the appropriateness of the parametrisations for crops within JULES and the input parameters for this one application. In particular, we attempt to test sections of the code in isolation wherever possible, to minimise the risk of tuning one parameter to compensate for a problem elsewhere in the model. This detailed dissection of the model is not something that has been previously carried out, and we believe that it is useful to understand

15    the model (both the crop parametrisation but also JULES vegetation in general) and also as a case study for setting up runs with other crops at other sites. Parameter settings for JULES are often inherited from one configuration to another, and a comprehensive re-examining of individual parameter settings against observations is rare. We intend this study to complement other work being carried out across the rest of the JULES community on crops and vegetation in general, such as improvements in the parametrisation of water stress.

20    We have submitted this study to GMD, since GMD encourages a wide variety of manuscript types, from the more traditional 'model description' papers (comprehensive presentations of models, with evaluation against standard benchmarks, observa-tions, and/or other model output), to more the specific categories, such as 'technical', 'development', 'model evaluation',

'Methods for assessment of models' and 'experiment description'. We feel that our study falls within the 'model evaluation paper' category, since it adds to the body of evaluation work on a pre-existing model. It is a particular strength of GMD that this category exists.

In this paper, we have included a discussion on the further work that would be needed before applying the tuned parameter values in this study to other sites or regions, since we felt this was desirable to place the results in the context of other JULES-crop development (and explicitly highlight important aspects not considered by this study, such as the response to water stress). However, in light of the referees' comments, we will de-emphasise this aspect of the paper and edit the text so that it is absolutely clear that we are not claiming that this work in any way 'completes' the validation of the model. We will also remove the discussion of the additional work needed before applying results derived on a site basis to a global run, as we think that introducing this issue has contributed to a lack of clarity in the aims of the paper.

Suggested change to manuscript: We have removed the sentence

The implications of our results on the choice of parameters and settings to be used in global runs of JULES-crop are also discussed.

from the abstract. We have changed

In this evaluation paper, we have looked in detail at how the input parameters can be tuned for one crop.

to

In this evaluation paper, we have looked in detail at how the input parameters in this pre-existing model can be tuned for one crop.

in the conclusion. We have also changed

We have also improved the parameters required in the crop-model part of JULES.

to

We have also improved the maize parameters required in the crop-model part of JULES.

and removed the sentence

This configuration would be applicable for a wider set of situations, for example, the ability to model the yield of non-irrigated crops would be a necessary requirement for addressing food security questions.

We have replaced the last paragraph

While this study has been carried out at one particular location, the JULES input parameters for individual crop tiles are designed so that they fully characterise a particular crop variety. Therefore, the logical next step, after addressing the issues described above, would be to test this model feature by investigating how well these new parameters perform at other sites which grow the same variety. To scale this up from the site level to global applications, it is important to consider that there

is a limit to the number of crop tiles that can be simulated in one run, due to data and computational resource constraints. Therefore each variety within a crop type cannot be simulated as a separate tile, with its own individual set of parameters. Previous JULES-crop global analyses, such as in Osborne et al. (2015), have therefore aimed at a compromise: four main crop types were considered (maize, soybean, wheat and rice), and each of these crop types were given spatially-varying thermal times between emergence and flowering and between flowering and harvest, which were tuned to observed growing season lengths. The other parameters were set to 'generic' parameter values for that crop type. In this way, some of the variation between varieties within a crop type was captured. The parameters given in Table 2 and Table 3 are a good starting point for a generic 'maize' tile, but would need to be tested at locations with different varieties, chosen for different climatic conditions, to determine whether they are able to sufficiently capture the characteristics of maize on a global scale to be able to address useful scientific questions. If not, the use of more than one 'maize' tile should be considered, or key parameters should be identified which could be allowed to vary spatially.

with one that focuses more on the value of this work as a case study:

While this study has focussed on modelling one crop variety at one site, it also provides a demonstration of how knowledge of the structure of the model can be used to tease apart different components of the model so that they can be tuned or evaluated against observations. This ranged from the tuning of parameters in simple allometric relations such as that relating stem carbon to canopy height, to tuning the canopy parameters using the external representation of the canopy scheme in `pySellersTwoStream`, up to running JULES with the crop model switched off and prescribed LAI and canopy height, in order to tune GPP without the complication of the feedback between GPP and LAI. It therefore provides a case study which can be followed when setting up and evaluating the model for other crop varieties and sites.

**2 Response to reviewer 1**

Crop phenology simulated by JULES-crop is still purely temperature driven. Many crop models have evolved to include impacts from other factors, such as precipitation, nutrient and day length. The authors should recognize biases it may bring in simulating crop phenology.

While these are important considerations for the model in general, these factors have a minimal effect on the results of this present study since the sowing, emergence, anthesis and harvest dates for each year are set using data from the site PI, and these are used to calculate the thermal times. As we mention in the conclusion, it would be interesting to investigate the performance of JULES at this site given generic values for the thermal times. This investigation would necessarily include consideration of other factors which could affect crop phenology but are not captured by the model.

Suggested change to manuscript: Put this point across more clearly in the conclusion by redrafting

Thirdly, these runs have been tightly constrained by observed sowing, emergence, flowering and harvest dates. For most regions, and for any climate projections, these sorts of data will not be available. Therefore, it would be useful to investigate the performance of the model in individual years at these sites when given generic values for the thermal time parameters.

so that it reads

Thirdly, these runs have been tightly constrained by using observed sowing, emergence, flowering and harvest dates to generate the thermal times needed as input to JULES. For most regions, and for any climate projections, this sort of data will not be available. Therefore, it would be a useful test of the model to investigate the performance at the Mead sites if the model is given generic values for the thermal time parameters.

Many of the results have not been well presented and discussed. For example, Figure 23 not only shows that the JULES-crop has low bias in simulating LAI, but also shows that the capability of the model to capture interannual variability of LAI is very limited. Why is that? The high bias of GPP and low bias of LAI is intriguing. This should be further explored and explained because it appears implying that models might get good results with wrong reasons.

The exploration of the GPP and LAI biases are key parts of the manuscript. We investigate the high bias of GPP in detail in section 4.1, where we prescribe LAI from the site observations, including investigating whether this is influenced by diffuse radiation fraction, air temperature, vapor pressure deficit or soil moisture (Fig 19 and Fig 20). We also discuss ways to address this problem, including changing the nitrogen distribution through the canopy, increasing $V_{\mathrm{cmax,norm}}$, decreasing alpha or changing the curvature parameters used to combine the limiting photosynthesis rates, and say why we decide that there is not a strong enough justification for these options. In our conclusion, we highlight the overestimation of GPP as one of the key results of the study.

Similarly, the underestimation of LAI is also highlighted in the conclusions as one of the key results, and vital in the interpretation of the model GPP when the crop model is switched on. In Section 4.2, we investigate this in more detail by re-running with different $\delta$ and $\gamma$ parameters, which give better agreement in the years identified as showing large LAI biases compared to the observations. We conclude that the issue is the high sensitivity of the crop in its early life (due to the positive feedback between LAI and GPP). We discuss the possibility of using a different parametrisation of SLA to improve things and we highlight the oversensitivity to initial conditions as one of the key findings of the study in the conclusions.

Suggested change to manuscript: Change

Including a decrease in leaf nitrogen concentration through the canopy would have the effect of making the light use of the plant more efficient, which would increase model GPP still further. Increasing $V_{\mathrm{cmax,norm}}$ would have the effect of decreasing model GPP at higher APAR values, but this would not solve the issue at mid-range APAR points $\sim 800\ \mu$mol photons $(\mathrm{m^2ground})^{-1}\mathrm{s}^{-1}$ and would also worsen the fit of the points with high diffuse radiation fractions.

25    to

Including a decrease in leaf nitrogen concentration through the canopy (while keeping the total amount of nitrogen constant) would have the effect of making the light use of the plant more efficient, which would increase model GPP still further. Decreasing $V_{\mathrm{cmax,norm}}$ would have the effect of decreasing model GPP at higher APAR values, but this would not solve the issue at mid-range APAR points $\sim 800\ \mu$mol photons $(\mathrm{m^2ground})^{-1}\mathrm{s}^{-1}$ and would also worsen the fit of the points with high diffuse radiation fractions.

to add a clarification and fix a mistake in the text.

The discussion of scaling up from sites to the globe is too superficial. At least, there are tests that the authors can do to facilitate this discussion. For example, the authors can compare the site simulations against global simulations (Osborne et al. 2015) or the site simulations with parameters from. Osborne et al., (2015). This will give us better impression on the uncertainties JULES-crop has now for global simulations. Otherwise, I did not see the reason why this is the conclusion of this study.

Suggested change to manuscript: As discussed above, we did not intend this to to come across as the conclusion of the study and we have removed this section from the conclusions.

It is also important to compare the efforts in JULES-crop with other land surface crop model, such as CLM-crop and ORCHIDEE-crop.

A comparison with other models is beyond the scope of the current study, as we focus on investigating whether the existing crop parametrisation within JULES can be used to model irrigated maize at one site.

Suggested change to manuscript: See changes suggested above, to clarify the aim of the study.

**3 Response to reviewer 2**

The central component of the paper is well written with a thorough description of the model and the parameter calibration, but the remainder of the manuscript lacks strong detail. For example, the introduction doesn't include much of a motivation for the study. What are the goals of the JULES crop model - yield, carbon, productivity and why is parameterizing for the Mead site a valuable exercise? The results and conclusion are also fairly brief.

Suggested change to manuscript: As discussed above, we have updated the manuscript to clarify the motivation of the study.

The authors mention in the abstract and introduction that they used observations from three MEAD FLUXNET sites, but they only use irrigated sites for model comparison? Why didn't they include the rainfed site in the analysis - especially when most cropland relies on rain to meet water demands?

While we agree that the ability of a crop model to reproduce rainfed sites is very important, this is beyond the scope of the current study, which focusses on irrigated maize. We have chosen to concentrate on an irrigated crop because when the plant becomes water stressed within JULES, this often dominate the carbon fluxes, and therefore the results become less useful for examining the other factors in e.g. the GPP calculation. Water stress for crops within JULES is parametrised in the same way as water stress for natural PFTs. It is likely that this parametrisation will change in the near future - there is a currently

a community-wide effort to improve the interaction between soil moisture and plant productivity and evapotranspiration in JULES, which will affect both crops and natural PFTS. In the conclusion of this manuscript, we state that the parametrisation of soil moisture stress of maize is the top priority for improving the representation of maize in general at this site.

In Section 2 and 3, some attention to the equations to define the parameters is needed to understand how the model works without reading other papers. The tables do not provide the necessary information to a non-user of JULES.

Suggested change to manuscript: We have edited the tables to include a brief text description of each parameter.

In Section 3, I would like some additional discussion on how the authors chose the parameters calibrated in this study. Was a sensitivity experiment done that indicated these parameters were important or were the parameters chosen in Tables 1-4 because they were convenient given the available observational data? If there was not a sensitivity study, perhaps the authors could highlight which parameters showed the most importance for the model results.

5      We have not carried out a systematic sensitivity study to the parameter values. While this approach can yield valuable information, we believe our approach is complementary to such studies and also has merit, since it improves the understanding of the structure of the model.

     Our parameter choice was based on site data where possible and on literature values otherwise. The choice of each individual parameter is discussed in section 3.3, 3.4 and 3.5. Where the choice of parameters is based on the site data, we provide the plot

10    to illustrate this, so that the reader can see the fit (Figures 1-16). Where the model is particularly sensitive to input parameters or the parameters are poorly constrained, this is discussed, for example, the nitrogen concentration per leaf area of the top leaf, the nitrogen distribution through the canopy and the specific leaf area at (or shortly after) emergence. In section 4.2, we do additional runs with two parameters varied: $\delta$ and $\gamma$, which are particularly important in the beginning of the run, and show that the results are very sensitive to these parameters. We highlight this as an issue with the model and include it as one of the

15    study's conclusions.

The authors compared the default JULES model without crops with the revised JULES crop model with updated parameters. This seems strange, what is the purpose of calibrating the LAI and height of JULES (without crops) with the newly parameterized JULES crop when a JULES-crop model already exists. I think it might also be more useful to compare the default JULES crop model from Osborne et al. (2015) with the newly revised site-specific parameterization. I also think it might be useful to look at the model performance at other sites.

     Our intention in including runs with the crop model switched off and prescribed LAI was not to show that the using the crop model is an improvement over not using the crop model. Rather, the runs with prescribed LAI were used to look at the simulation of GPP with the feedback between GPP and LAI switched off. This allows a cleaner validation of the parameters and parametrisations used in calculating model GPP. Running the model at other sites is beyond the scope of this current study.

Suggested change to manuscript: Clarify this point by adding the extra paragraph to the conclusions described above, which emphasises the use of the prescribed LAI runs to test components of the GPP calculation. Change the paragraph

5        In previous analyses with JULES-crop, it has been assumed that model photosynthesis and respiration parameters can be set to the default C3 grass values for C3 crops and the default C4 grass values for C4 crops. We have shown that a significant improvement can be made when modelling irrigated maize if these parameters are tuned to results from the literature for maize. We have also improved the maize parameters required in the crop-model part of JULES (such as partition fractions and allometric constants) by tuning directly to observations.

to

10        In previous analyses with JULES-crop, it has been assumed that model photosynthesis and respiration parameters can be set to the default C3 grass values for C3 crops and the default C4 grass values for C4 crops. We have used literature results and the observations available at this site to improve the maize parameters required in both the crop-model part of JULES (such as partition fractions and allometric constants) and the generic vegetation code.

It would also be useful to perform an uncertainty analysis of the parameters. This would be a valuable not only for the current model analysis, but also for extrapolating to other sites or globally. I'm not asking the authors to do this for this publication, but parameter uncertainty should be included in the discussion.

See response to the comment on a 'sensitivity study' above.

Eq. 1 has several parameters for temperature that aren't clearly defined (Table 3?). Relatedly, the second column in Tables 1-4 is not particularly useful to the reader; perhaps more appropriate would be a description for the variable rather than the model assigned parameter name.

15        These temperature parameters are the parameters in a triangular function, as defined by equation 1 and line 10-11 on p3. We will add an extra column to the tables to give a text summary of each parameter. The second column is important for users of JULES, as it removes any ambiguity between the parameters we define in the paper and the JULES name list variables.
        Suggested change to manuscript: New column added to tables to give a description of the JULES namelist variable in words.

Eq. 3: What is the difference between j and i? Again, it's not clear to a non-user of the model what those parameters are since they are not defined in the text or in the table.

        j is defined in line 23 and i is defined in line 21. They are indices representing the four carbon pools. In equation 3, i is a
20   particular instance of a carbon pool, whereas j is the summation index.

P 35, L 19-20: Figure 27 should be referenced here.

Thanks for spotting this omission - we have added this to the new manuscript.

Figure 26 & Figure 27: Why does US-Ne2 2010 have no observations?

This was missing because it was not part of the original observational dataset considered in the analysis. However, we have been able to source this missing data and have updated Figure 26 and 27 to include the lines for US-Ne2 observations.

Suggested change to manuscript: New data added to plots in Figure 26 and Figure 27.

5 ## 4 Response to reviewer 3

Although the Abstract indicates that observations for maize at all three sites of Mead, Nebraska including one rain fed site (i.e. US-NE3) has been considered in deriving model parameters, according to the Introduction and the other sections, only the model performance for irrigated maize has been evaluated; some results from the rain fed maize might enhance the quality and validity of this study.

As discussed above, this study looks specifically at modelling irrigated maize at this site. Evaluating the performance of the model at the rainfed site would be beyond the scope of this current work and in the conclusion, we recommend that work on the parametrisation of soil moisture stress of maize should be the top priority for improving the representation of maize at this site.

10 Suggested change to manuscript: See description of how the manuscript has been changed to clarify the aims of the study.

The JULES-crop simulations were run on crop tiles. At the US-NE2 site, soybean crop is also present in crop rotation on the same crop tile during even-numbered years. It would have been interesting to see how the model simulates crop-rotation on the same crop tile/s, as the paper also mentions that the model has been differently parameterized for C3 and C4 crops. This is very important, especially as the authors have plans for coupled-runs in the future, where the model should be able to simulate the carbon fluxes over a continuous time series. At least some mention/description on the model performance with regard to crop rotation needs to be included.

This particular study is concerned solely with assessing how well JULES-crop is able to model irrigated maize at this site - looking at the performance of soybean would be beyond the scope of this study.

There is an intention in the JULES community that JULES-crop will eventually be available for use in coupled model runs. However, a considerable amount of model development would be needed before this would be even technically possible. Once

15 it is technically possible, there would still be large amount of work to test the performance of the crop parametrisation within

the coupled system. We have not intended to give the impression that the work done in this paper shows that the model is ready for such runs, and we have updated the manuscript to remove this ambiguity.

Other work within the JULES community is currently focussing on the modelling of crop rotation for the case of rice and wheat in India and Bangladesh, but this is beyond the scope of this study.

5      Suggested change to manuscript: See above for the changes in describing the purpose of the study, in particular taking out the discussion relating to global runs.

The paper is a bit too long; especially the number of figures is too large. Please try to reduce the number of figures, leaving only those that are essential and directly related to the predicted fluxes.

After reading this comment, we have carefully gone through each figure to determine whether it is really necessary to have it in the main body of the paper, and feel that they are all required to fulfil the aims of the paper. As discussed above, we are attempting to pull apart the pieces of the model and see how suitable the parameters and parametrisations are for this particular
10      dataset and provide a case study of how to set runs up for other crops and sites. Therefore, we feel it is valuable to present this process it its entirety, and that the plots used to tune the parameters and test individual model components are, if anything, more useful than the final result for GPP with everything switched on.

Suggested change to manuscript: We have added a section to the conclusion to describe the use of this study as a case study for setting up the model at other locations and for other crops.

Although the text on line 13-14 on p 35 mentions that the use of certain parameters improves the prediction of LAI, the improvement in the magnitude can be seen only during certain years when we compare the Figures 23 and 24. It seems that the model still needs improvement with regard to LAI, as the seasonality is not properly captured by the modeled LAI compared to the observed LAI.

15      We agree that the model is not properly capturing LAI in all years, which we conclude is due to an oversensitivity to initial conditions, and this is one of the key findings that we highlight in the conclusions. As discussed above, this study is looking at a particular application of a pre-existing model, and an important part of this task is pointing out where the model needs further development to perform well for this use case.

The reader hardly can get any information from Tables 1-4; an additional column which describes each parameter listed in columns 1 and 2 might be helpful (or the model terms in column 2 could be replaced with easily understandable descriptors of each parameter). The column heading of the last column may be changed to 'Remarks' (instead of 'Discussion').

Suggested change to manuscript: We will add an extra column giving a text summary of each parameter. We will change the
20      heading of the last column from 'Discussion' to 'Remarks'.

The paper does not seem to be in its final form, as still there are some typos and other errors, some of which are described below. So a thorough check on those is also needed.

P3 L13 'resp'should be replaced with 'respectively'

Suggested change to manuscript: We have removed this abbreviation.

P5 L20 has 'a number options'. Please correct it.

Suggested change to manuscript: We have changed 'a number options' to 'a number of options'.

P5 L24 Zenith angle dependence (of what)?

In the sentence "The equations for absorption and scattering at each layer for the incident diffuse beam and the incident direct beam (including the zenith angle dependence) are solved separately, taking into account the distribution of leaf angles.",
5  the 'zenith angle dependence' is in the absorption and scattering at each layer.

Suggested change to manuscript: Reorder the sentence to remove this ambiguity i.e. change it to

> The equations for absorption and scattering at each layer for the incident diffuse beam and the incident direct beam are solved separately, taking into account the distribution of leaf angles and the zenith angle.

P6 L17 Q10 should be replaced with Q10

10  Suggested change to manuscript: We have replaced this occurrence of Q10 with $Q_{10}$.

P14 L11 Instead of 'downloaded 15.09.2016', please provide a proper reference/web source.

Suggested change to manuscript: We have replaced 'downloaded 15.09.2016' with a reference to the code availability section, which already contains the link to the source. We have added the sentence 'The version used in this study was downloaded on 15.09.2016' to the code availability section.

P15 L24 (and everywhere else) 'Parametrisation' needs to be replaced with 'parameterization'.

We have checked with the Copernicus editorial team, and 'parametrisation' is an accepted spelling variant. We have made
15  sure that this spelling is consistent throughout the document.

P 20 Figure 4 has several lines in each color. Unless the authors explain what those are, the figure does not have much meaning to it (e.g. what are those several lines in black color mean? Which site does each of those correspond to?).

This information is already included in the caption of Figure 4, which reads

Green leaf biomass against DVI. Blue, green, red lines are derived from US-Ne1, US-Ne2 and US-Ne3 observations respectively. Black lines are generated using model parameters from Osborne et al. (2015) (left plot, solid lines) and the new, tuned parameters (right plot, dashed lines).

P 47 Heading of Table 5 mentions 'thermal units in degree days', whereas 'degree days' does not appear anywhere else in the text (According to P3 L1 crop development status is parameterized by a crop development index (DVI) which is determined by specific thermal time parameters set by the user (P3 L6-7). Degree days in Table 5 need to be related to the above description on p3.

[revised manuscript text omitted]
_{\text{cmax}} & \text{for} \quad I_{\text{APAR}} \Delta \text{LAI} > 10 \mu \text{mol} \, CO_2 (m^2 \text{ground})^{-1} s^{-1} \\ f_{dr} V_{\text{cmax}} & \text{otherwise} \end{cases} \tag{20}$$

to allow for the inhibition of dark respiration during daylight. $R_d$ is summed over the canopy levels for sunlit and shaded leaves to get $R_{dc}$, the canopy dark respiration in (in mol $CO_2$ $(m^2$ ground$)^{-1}$ $s^{-1}$).

The plant maintenance respiration in kg C $(m^2$ ground$)^{-1}$ $s^{-1}$ is calculated (for the setting `l_scale_resp_pm=T`) using

$$
\begin{aligned}
R_{pm} &= 0.012 R_{dc} \beta \left( 1 + \frac{N_{\text{root}}}{N_{\text{leaf}}} + \frac{N_{\text{stem}}}{N_{\text{leaf}}} \right) \\
&= 0.012 R_{dc} \beta \left( 1 + \mu_{rl} \frac{C_{\text{root}}}{C_{\text{leaf}}} + \mu_{sl} \frac{C_{\text{stem}}}{C_{\text{leaf}}} \right),
\end{aligned}
\tag{21} \tag{22}
$$

[revised manuscript text omitted]

---

## Author Response (AR2)

**Response to reviewers: Evaluation of JULES-crop performance against site observations of irrigated maize from Mead, Nebraska**

[revised manuscript text omitted]

there are a few similar parameterization efforts made by land surface modellers, many of which even published in GMD. For the convenience of readers, it is necessary to put these progresses in context.

**Suggested changes to manuscript:**

We have added this paragraph to the introduction

Other land surface models include specific representations of key crops. CLM-crop has been evaluated at the site level for several crop types (maize, soybean and spring wheat (Drewniak, 2013); winter wheat (Lu et al, 2016)) and physiology parameters were calibrated to optimize productivity (Bilionis et al, 2015). Globally, the model is being developed to included transient land cover and land use for the Land Use Model Intercomparison Project (LUMIP) contribution to CMIP6 (Lawrence et al, 2016). ORCHIDEE-CROP has been evaluated for maize and winter wheat at a number of European sites (Wu et al, 2016) and was shown to reproduce the seasonality of leaf area index and carbon and energy fluxes. Similarly, the incorporation of a phenology scheme into the SImple Biosphere (SIB) model improved the prediction of both leaf area index and carbon fluxes for maize, soybean, and wheat crops at a number of sites in North America (Lokupitiya et al, 2009). Song et al (2013) implemented crop-specific phenology and carbon allocation schemes into the Integrated Science Assessment Model (ISAM) land surface model and calibrated against observational data from a corn-soybean rotation at Mead and Bondville (US) sites. This model was able to reproduce the diurnal and seasonal variability of carbon, water and energy fluxes.

**2 Response to referee 2**

It would benefit the reader if the authors did a better job framing the manuscript with the introduction. The introduction

highlights previous work with JULES-crop that could be improved with better crop parameters (P2, L 12-13 and L18-19). But the paper also focused on the non-crop version of JULES and how driving the model with prescribed LAI and height could help tease out feedbacks on GPP. In fact, that goal doesn't become clear until the results section. The other goal that becomes clear late in the manuscript is that this study is really presenting a methodology to tune parameters at other sites and understand possible structural uncertainty within JULES, not necessarily to improve parameters for use in JULES-crop. The authors claim they have clarified the aim of the manuscript, but I see no evidence of this until the end of the conclusions. The introduction needs to clearly set up the problem that is being addressed by this manuscript.

**Suggested change to the manuscript:**

In the introduction, change

In this model evaluation paper, we use the observations available at the Mead FLUXNET sites US-Ne1, US-Ne2 and US-Ne3 to investigate how well each individual component of JULES performs for maize and how much of an improvement can be

5 achieved by using more appropriate parameter values, taking into account advances in the JULES code since the Osborne et al. (2015) study. We will use these new sets of parameters in JULES-crop runs for irrigated maize at Mead.

to

In this model evaluation paper, we use the observations available at the Mead FLUXNET sites US-Ne1, US-Ne2 and US-Ne3 to investigate how well each individual component of JULES performs for maize and how much of an improvement can be

10 achieved by using more appropriate parameter values, taking into account advances in the JULES code since the Osborne et al. (2015) study. **This investigation splits into three distinct parts. We initially look at which processes and parameters can be tuned directly to maize observations from the Mead sites, without running the model. Secondly, for parts of the code shared between natural PFTs and crops in the model (the calculation of gross primary productivity and respiration), we test the performance of the tuned parameters by running JULES with the crop model switched off and forcing**

15 **with observed leaf area index (LAI) and canopy height, to remove the feedback between net primary productivity and LAI. Finally, we will use the tuned parameters in JULES runs for irrigated maize at Mead with the JULES-crop parametrisation switched on.**

In Section 3.2 (Model setup), I suggest the authors indicate that the 1st run includes the crop parameters earlier than at the end of the section; otherwise it appears that maize is being run as a natural PFT.

As described in the Model set-up section there are two types of JULES runs used in this paper. In the first type, maize is being run as a natural PFT, with LAI and crop height constrained to be the same as the observations. In the second type, the

20 JULES-crop parametrisation is switched on.

**Suggested change to the manuscript:** Hopefully the text added to the introduction (above) also clarifies this second point.

> If testing the model performance at the rain fed site is outside the scope of the study, why is the site used for parameter estimation? It should be eliminated from the manuscript to focus on the irrigated sites only.

We would like to keep observations from the rainfed site in some plots in the parameter tuning sections. In many cases, the processes and parameters that can be tuned outside of the rest of JULES are independent of the JULES water stress parameter (which multiplies GPP and/or NPP and is the only way that plants in JULES are affected by switching the irrigation on). For example, the allometric relation in the model between crop height and stem biomass is not influenced by the water stress parameter. In addition, for tuning the canopy properties, we also used DVI up until flowering only, and the observations did not show significant water stress at the rainfed site during this time. It therefore makes sense to consider observations from all maize sites at Mead - irrigated and rainfed - when tuning these parameters. For some plots, e.g. Fig 15, removing the rainfed observations removes a significant amount of the available observations. In reality, there are more ways that a crop can be affected by water stress than is captured by the model e.g. the partition fractions would change to prioritise root growth. Therefore, in all plots where the rainfed sites contribute, the rainfed observations are distinguished by the line colour or the symbol type. In each case, we were able to confirm whether the observations from the rainfed site were consistent with the observations from the irrigated sites. We therefore decided to use rainfed observations throughout the parameter tuning section. There is one exception to this: Figure 8, since we would expect the initialisation to be dependent on planting density, which is different in the irrigated and rainfed fields.

However, as a result of the referee comments, we have now removed all rainfed observations from the plots in the Results section. This has meant removing the rainfed data from the two plots in Figure 19 and and the four plots in Figure 20. We had included these data points in the previous versions of the manuscript because only data for LAI 3.5-4.5 before flowering was considered, and we checked that there was no significant influence of soil water stress on these results. However, since the purpose of this section is to investigate the model runs for the irrigated sites, it makes sense to edit Fig 19 (right) so that it only includes model results for the irrigated sites (we had actually used a toy model to mimic JULES at both the irrigated and rainfed sites), in which case we also need to remove the rainfed observations in Fig 19 (left) and Fig 20. Because these plots use hourly observations there is a sufficient quantity of data that the conclusions drawn from these plots in the text are still valid. The text has been changed accordingly.

Note that Figure 21 already only used observations from the irrigated sites, because the values are derived from season totals of NPP and GPP, which are strongly affected by water stress.

**Suggested change to the manuscript:** We have redone 6 plots to remove data for the rainfed site. In the observations section, we have added

> Observations from all three sites were considered in the input parameter tuning, whereas only observations from the irrigated sites were used to drive and validate the model runs.

We have added this text to the model setup section:

> Observations from both the irrigated sites at Mead and the rainfed site at Mead were considered when tuning the model input parameters that were designed to take the same value whether irrigation is switched on or off in the model. However, in

these cases, observations from the rainfed site are clearly denoted on the plots, in order to check for cases where these model approximations break down.

In the carbon partitioning section, we have added:

> In addition, in reality, water stress can also increase the fraction of NPP going to the roots (see discussion in e.g. de Vries et al. (1989) and Song et al. (2013) ), but this effect is not taken into account in JULES-crop. **However, we do not see a notable difference between the irrigated sites US-Ne1 and US-Ne2 (blue and green lines respectively) and rainfed site US-Ne3 (red lines) in Figure 1.**

We have also added this text to the caption of Fig 14 (this was omitted by accident from the earlier draft):

> and all data are between emergence (DVI=0) and flowering (DVI=1)

Sometimes the variables in an equation are not defined in the text and it is up to the user to search for them in the tables (for example, the To, Tb and Tm in equation 1, $\alpha$ and $\beta$ in equation 3, $\mu$ and $\nu$ in equation 4). Please go through all the equations and make sure they are well described for the reader in the text.

The parameters given as examples here (To, Tb, Tm, $\beta$, $\mu$ and $\nu$) are those defined by the equations they appear in and have no physical interpretation outside these equations. We will change the manuscript to make this more explicit.

**Suggested change to manuscript:** We have changed

> effective temperature is defined by ... i.e. a triangular function, peaking at $T = T_o$, which is zero below $T = T_b$ and above $T = T_m$. $T_o$, $T_b$ and $T_m$ are parameters specified by the user for each crop. $T_o$, $T_b$ and $T_m$ are given in Kelvin and thermal time in units of degree days.

to

> effective temperature is defined by ... i.e. a triangular function, peaking at **an optimal temperature** $T_o$, which is zero below **a base temperature** $T_b$ and above **a maximum temperature** $T_m$. $T_o$, $T_b$ and $T_m$ are parameters specified by the user for each crop. $T_o$, $T_b$ and $T_m$ are given in Kelvin and thermal time in units of degree days.

We have changed

> $\alpha_{\mathrm{harv}}$ and $\beta_{\mathrm{harv}}$ are both set to zero. All other $\alpha_i$ and $\beta_i$ are set by the user for each crop. Note that $\sum_j p_j = 1$.

to

> $\alpha_i$ **and** $\beta_i$ **are numerical constants that are tuned to observational data**. $\alpha_{\mathrm{harv}}$ and $\beta_{\mathrm{harv}}$ are both set to zero. All other $\alpha_i$ and $\beta_i$ are set by the user for each crop. Note that $\sum_j p_j = 1$.

We have added

> $\nu$ and $\mu$ are numerical constants that are tuned to observational data.

**3 Response to referee 3**

The main point of referee #3 was that the paper would contain unnecessary detail that could be easily found in the cited literature. One suggestion was to make section 2 more concise, e.g. by compiling a table with the parameters described in the text with a column for the original JULES parameter, one for the JULES-crop parameter (if that distinction is helpful), one for the parameter used in your study and one column with remarks and the relevant equation.

We find it difficult to see how we can reduce the contents of the model description without losing information that we later refer to in the experimental set-up section. The two published articles which contain some of this information are Clark et al (2011) (parameterisations used for both natural PFT and crop tiles) and Osborne et al. (2015) (parameterisations that just get used for crops). However, in both cases, the experimental set-up section relies on details that have not been covered by these papers. Furthermore, the JULES model (a community model) has had many features added since these two publications (e.g. all parameters with asterisks in table 1, table 2 and table 3). To our knowledge, with the exception of the `l_trait_phys` option (which we set to false here anyway, so that we default to the 'old' behaviour), these have not been described in any published literature. We feel that it is essential to have all relevant parts of the model clearly defined somewhere in the manuscript if we are to unambiguously tune the parameters in Section 3, and there would not be sufficient space in the existing tables†, nor the possibility of showing how different sections of the model fit together. This is needed for the discussion on which parts of the model can be picked apart and tested in isolation and which parts of the model can only be tested within the rest of the model.

† Table 1, Table 2, Table 3 and Table 4 include the notation used in the JULES namelists, the notation used in this paper, the parameter values used in the Osborne et al. 2015 study, the values used in this study and a brief text summary of the parameter, including the section where the discussion of this parameter can be found in the text. These tables already cover 5 1/2 pages at present (in the GMDD manuscript format).

**Suggested change to manuscript:** Given the concerns of the referee, we have moved this section to the appendix in this version of the manuscript, so that the length of this section does not affect the flow of the paper.

Another point of referee #3 was that figures 3 and 4 are too busy. I would recommend to also indicate what site was modeled by the same colors as used for the observations in figure 4 and to use dashed lines for modeled values in both panels. The distinction between the Osborne parameterization and yours is already given by the two panels.

**Suggested change to manuscript:** We would prefer to keep the convention of using solid lines using Osborne et al 2015 parameters and dashed lines using new parameters if possible, to keep consistent with the other figures in experimental set-up section of the manuscript. We have attempted to improve the clarity of Figure 4 by increasing it from two panels (left: obs and 'old' model and parameters, right: obs and 'new' model and parameters) to three panels (left: obs, middle: 'old' model and parameters, right: 'new' model and parameters) and normalising the y axis, and have updated the text accordingly.

[revised manuscript text omitted]